# Traffic light optimization with low penetration rate vehicle trajectory data

Xingmin Wang [1], Zachary Jerome [1], Zihao Wang [1], Chenhao Zhang[2], Shengyin Shen [3], Vivek Vijaya Kumar[4], Fan Bai[4], Paul Krajewski[4], Danielle Deneau[5], Ahmad Jawad[5], Rachel Jones[5], Gary Piotrowicz[5] & Henry X. Liu [1,3,6] ✉

Traffic light optimization is known to be a cost-effective method for reducing congestion and energy consumption in urban areas without changing physical road infrastructure. However, due to the high installation and maintenance costs of vehicle detectors, most intersections are controlled by fixed-time traffic signals that are not regularly optimized. To alleviate traffic congestion at intersections, we present a large-scale traffic signal re-timing system that uses a small percentage of vehicle trajectories as the only input without reliance on any detectors. We develop the probabilistic time-space diagram, which establishes the connection between a stochastic point-queue model and vehicle trajectories under the proposed Newellian coordinates. This model enables us to reconstruct the recurrent spatial-temporal traffic state by aggregating sufficient historical data. Optimization algorithms are then developed to update traffic signal parameters for intersections with optimality gaps. A real-world citywide test of the system was conducted in Birmingham, Michigan, and demonstrated that it decreased the delay and number of stops at signalized intersections by up to 20% and 30%, respectively. This system provides a scalable, sustainable, and efficient solution to traffic light optimization and can potentially be applied to every fixed-time signalized intersection in the world.

There are more than 320,000 signalized intersections in the United States (US). Annually, drivers experience roughly $22.9 billion in direct and indirect congestion costs at these intersections[1]. Much of these costs are the result of outdated or improper traffic signal operations, which the 2019 National Traffic Signal Report Card gave a C+ grade[1]. Traffic signal retiming is widely regarded by traffic engineers as one of the most cost-effective methods for reducing congestion and energy consumption in urban areas as it doesn't require any major changes to the existing infrastructure[2–5]. However, the high installation and maintenance costs of vehicle detectors have prevented the

widespread implementation of detector-based systems such as vehicle-actuated control and adaptive traffic control systems (ATCS)[6–8]. As a result, a large proportion of the signalized intersections in the US do not have detection capabilities and are still controlled by fixed-time traffic signals[1,7]. Signal retiming at these intersections still relies on manual data collections and is therefore only executed every 3–5 years in practice[9]. As traffic demand undergoes natural changes or growth, these timing plans become outdated, which increases congestion and energy costs. Similar situations can be observed around the world.

[1]Department of Civil and Environmental Engineering, University of Michigan, Ann Arbor, MI 48105, USA. [2]Department of Computer Science and Engineering, University of Michigan, Ann Arbor, MI 48105, USA. [3]University of Michigan Transportation Research Institute, Ann Arbor, MI 48105, USA. [4]General Motors Research and Development, Warren, MI 48092, USA. [5]Road Commission for Oakland County, Beverly Hills, MI 48025, USA. [6]Mcity, University of Michigan, Ann Arbor, MI 48105, USA. ✉e-mail: henryliu@umich.edu

In recent years, vehicle trajectory data has become increasingly available from various connected vehicle services such as en-route navigation, roadside assistance, and ride-hailing services. Monitoring traffic through vehicle trajectory data offers many advantages over fixed-location detectors and sensors[10-12]. It has a much larger coverage area than detector data because it is available at almost every intersection, especially those with higher traffic volumes (Fig. 1a, Supplementary Movie 1). While detector data can only provide traffic counts and estimated speeds at certain locations, vehicle trajectory data spans the entire spatial-temporal space and provides more enriched information such as delay, number of stops, and travel path (Fig. 1b). This presents an unprecedented opportunity for traffic signal optimization that can reduce traffic congestion without additional sensor instrumentation on physical road infrastructure.

This paper focuses on optimizing fixed-time traffic signals using connected vehicle trajectories without relying upon any road-side detectors (e.g., loop detectors, cameras). Although many existing studies have investigated traffic signal control with connected and automated vehicles (CAV)[13-19], they assume a high penetration rate (i.e., the proportion of CAVs to the overall number of vehicles), which is not realistic in the current practice. In this study, we aim at optimizing traffic signals utilizing vehicle trajectories at the currently available market penetration rate. In this case, one major challenge is the sparse and incomplete observation of the overall traffic state. Some studies have developed statistical methods to estimate certain traffic flow parameters such as traffic volumes or queue lengths[20-24], but they can only be used for traffic monitoring purposes due to the lack of an explicit traffic flow model. For traffic signal optimization, it is important to have the capability to predict traffic flow performance under different traffic signal parameters.

Stochastic traffic flow models can be used to estimate and predict the overall traffic state from incomplete observations. However, most existing traffic flow models do not fit with vehicle trajectory observations. Eulerian and Lagrangian coordinates are the two most used coordinate systems in existing traffic flow models (Fig. 1c). Eulerian coordinates split the spatial-temporal space into grids and define the traffic state as the density in each grid. Trajectory data does not provide measurements in Eulerian coordinates and hence cannot be directly used to calibrate the traditional Lighthill-Whitham-Richards (LWR) model and its variants[25-31]. Vehicle trajectory data is in the form of Lagrangian coordinates which keep track of the vehicle's movement, but traffic flow models under Lagrangian coordinates suffer from high dimensionality and are not applicable to large-scale applications. In addition, models utilizing both Eulerian and Lagrangian coordinates become more complicated at higher dimensions when extended to stochastic settings[30-34]. Due to the lack of a suitable traffic flow model based on vehicle trajectory data, only a handful of studies have used such data to attempt traffic signal optimization heuristically with a very limited scope for implementation[35-37]. For a more comprehensive review of related works, please refer to Supplementary Section 2.

In this paper, we propose a stochastic traffic flow model under Newellian coordinates, which is established based on Newell's car

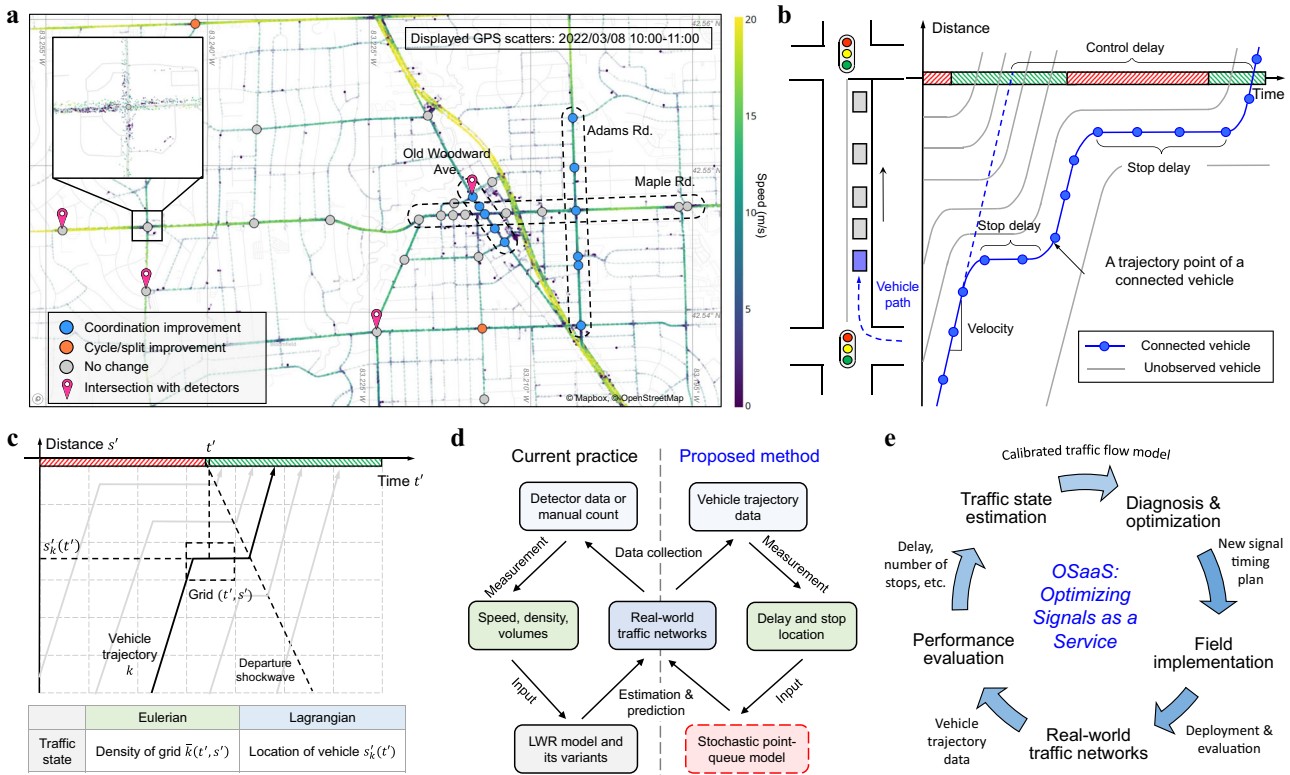

**Fig. 1 | Traffic signal retiming with vehicle trajectories. a** Trajectory point scatters in the City of Birmingham, Michigan, which has a total of 34 signalized intersections including three main corridors and some isolated intersections. Each point represents a vehicle's location at a certain timestamp and the color of the point changes with speed. Corridors and intersections that were identified with traffic signal re-timing opportunities are labeled with different colors. **b** Time-space (TS) diagram and connected vehicle trajectory. The blue line shows the vehicle trajectory of a connected vehicle. Each blue dot represents a trajectory point. **c** Eulerian and Lagrangian traffic state representations. Eulerian traffic state representation defines the traffic state as the density of each cell while the Lagrangian keeps track of the movement of the vehicle. **d** Comparison of current practice based on fixed-location detector data and the proposed method with vehicle trajectories. LWR models are frequently used to model traffic flow from detector data which directly provides speed, density, and volumes at certain locations. We utilize the stochastic point-queue model under the Newellian coordinates where vehicle trajectory data, which directly provides delays and stop locations, is the only input. **e** OSaaS system: an integrated closed-loop system with performance evaluation, traffic state & parameter estimation, diagnosis, optimization, and field implementation.

following model[38]. We show that a simple point-queue model under the proposed Newellian coordinates can sufficiently capture spatial-temporal traffic state through the probabilistic time-space (PTS) diagram. This simplification is made feasible by ignoring stochastic and heterogeneous driving behavior since most of the system uncertainty arises from the stochastic traffic demand as well as sparse observations at low penetration rates. The main advantage of the proposed model is that it is a stochastic model with much lower dimensions and can be directly calibrated by taking the vehicle trajectory data as the input. It enables us to apply different estimation algorithms to estimate unknown traffic states and parameters. We demonstrate that, even at a low penetration rate, recurrent traffic states can be accurately reconstructed by aggregating sufficient historical data. This enables us to develop a traffic signal optimization method that can transform the state-of-the-practice at scale (Fig. 1d).

With the proposed methods, we present a large-scale traffic signal optimization system (OSaaS: Optimizing Signals as a Service) based on vehicle trajectory data collected by connected vehicle service providers. OSaaS is a closed-loop signal optimization system that includes monitoring, modeling, diagnosis, and optimization (Fig. 1e). In each retiming iteration, delay and stop measurements are first calculated from the collected trajectories to evaluate traffic performance. Traffic flow parameters such as the penetration rate and arrival rate are then estimated based on the proposed traffic flow model. Based on the calibrated model, the diagnosis module finds the traffic performance optimality gap with respect to different signal timing parameters, which indicate different traffic signal re-timing opportunities. Optimization algorithms are developed to update signal timing parameters for intersections that show potential for improvement. In this way, the OSaaS system can dynamically optimize traffic signal periodically every few weeks, compared to the 3-5 years in the current practice.

With vehicle trajectory data from General Motors (GM), the system was tested in the City of Birmingham, Michigan through a field implementation in March 2022. This included citywide monitoring, modeling, diagnosis, and optimization of all 34 signalized intersections in the city. Most of these signalized intersections are not equipped with any vehicle detectors, so the proposed system provided previously unavailable opportunities. Implementation of the new timing plans resulted in significant reductions in control delay and the number of stops. By utilizing the trajectory data as the only input and not requiring any additional infrastructure, OSaaS provides a more scalable and economical solution to traffic signal retiming which can potentially be applied to every fixed-time traffic signal in the world.

## Results
### Newellian coordinates, stochastic point-queue model, and probabilistic time-space diagram

The Newellian coordinates are established on the assumption that all vehicles follow a homogeneous deterministic Newell's car-following model[38]. This assumption holds for urban traffic featured by interrupted flow, where stop-and-go is the dominant characteristic of vehicle trajectories and vehicle delay mainly comes from the stopping time caused by traffic signals and queueing. When the penetration rate is low, most of the uncertainty arises from incomplete observation and stochastic traffic demand. Therefore, we ignore the heterogeneity and stochasticity of driving behaviors in this paper. We also apply a discrete approximation: for each time interval $\Delta t$, traffic flow comes in binary with either 0 or $\Delta u$. The unit traffic flow $\Delta u$ is determined by the number of vehicles that comes at the saturation flow rate within time interval $\Delta t$. More details about the discrete approximation are available in the Methods section. Please also refer to Supplementary Section 1 for the table with notations used in the paper.

The distorted grid in Fig. 2a is an illustration of the proposed Newellian coordinates. The horizontal and vertical intervals are the time interval $\Delta t$ and the jam space headway $h$, respectively. The slope

of vertical axis is the free-flow speed $v_f$. The Newellian coordinates of each vertex in the grid is determined by $(t,n)$: the horizontal axis $t$ represents the free-flow arrival time (in units of $\Delta t$), which can be interpreted as the time when a vehicle would have arrived at the intersection if it traveled at the free-flow speed and did not have to slow down or stop because of background traffic or the traffic signal; the vertical axis $n$ denotes the number of unit traffic flows (in units of $\Delta u$), which directly corresponds to the stopping location of $n$th unit traffic flow.

The transformation between the real-world time space coordinates $(t',s')$ and Newellian coordinates $(t,n)$ is given by:

$$\begin{cases} t' = t - \frac{n \cdot h}{v_f} \\ s' = n \cdot h \end{cases} \tag{1}$$

The major difference is that Newellian coordinates use the free-flow arrival time as time $t$ instead of the actual real-world time $t'$. Based on the previously introduced assumption and discrete approximation, vehicles only travel on the edges of the Newellian coordinates. By taking trajectory $k$ in Fig. 2b as an example, it can be encoded as $(a^k, x^k, b^k)$ where $a^k$ is the free-flow arrival time, $x^k$ is the stop location, and $b^k$ is the departure time when it leaves the intersection. The difference between the departure time and free-flow arrival time $|b^k - a^k|$ is the control delay[39].

Newellian coordinates enable us to convert all vehicle trajectories to a point-queue representation (Fig. 2b). Let $X^n(t)$ represent the spatial queue length (in units of $\Delta u$), which corresponds to the location of the last stopped vehicle at the end of time $t$. Let $X(t)$ denote the number of stopped vehicles at time $t$. Based on the deterministic driving behavior assumption, they have a deterministic mapping relation: $X(t)$ equals the $X^n(t)$ minus the elapsed green time $t - t^r$, where $t^r$ is the end of the red time (Fig. 2b). It is easy to verify that the dynamics of $X(t)$ is given by:

$$X(t) = X(t-1) + A(t) - B(t) \tag{2}$$

where $A(t)$ and $B(t)$ denote the arrival and departure, respectively. Both $A(t)$ and $B(t)$ are binary according to the discrete approximation. Following simple dynamics given by Eq. (2), $X(t)$ is called a point queue since it does not have spatial information. With both queue lengths $X(t)$ and $X^n(t)$, the point queue $X(t)$ is used as the main representation for the traffic state in Newellian coordinates since it has much simpler dynamics; the spatial queue $X^n(t)$ can be derived from the point queue $X(t)$ whenever it is needed. Supplementary Movie 2 provides further illustrations of the proposed Newellian coordinates and point-queue representation.

Due to the uncertainty caused by the incomplete observation and stochastic traffic demand, a stochastic model is required. The deterministic point-queue model can be easily converted to a stochastic version (i.e., a stochastic queueing model) by applying a stochastic arrival process. It is assumed that the vehicle arrival $A(t)$ at each time follows a Bernoulli distribution with arrival probability $a(t)$. There is a departure, i.e., $B(t) = 1$, whenever the traffic light is green, and the existing queue is not empty. In this way, we have specified the transition of the stochastic queueing model, and the queue length distribution can be derived given the input arrival and traffic signal state (table in Fig. 2c). Although stochastic queueing models have been widely studied to model urban traffic networks[40–44], few have established their connection with vehicle trajectory data[45].

Figure 2c shows how the stochastic point-queue model can be projected back to the spatial-temporal space using the probabilistic time-space (PTS) diagram. As aforementioned, vehicles only travel on the edges of the Newellian coordinates. Let $\rho^n(t,n)$ and $\rho^t(t,n)$ denote the probability that there are vehicles traveling on the vertical and horizontal edges, corresponding to the free-flow and stop states,

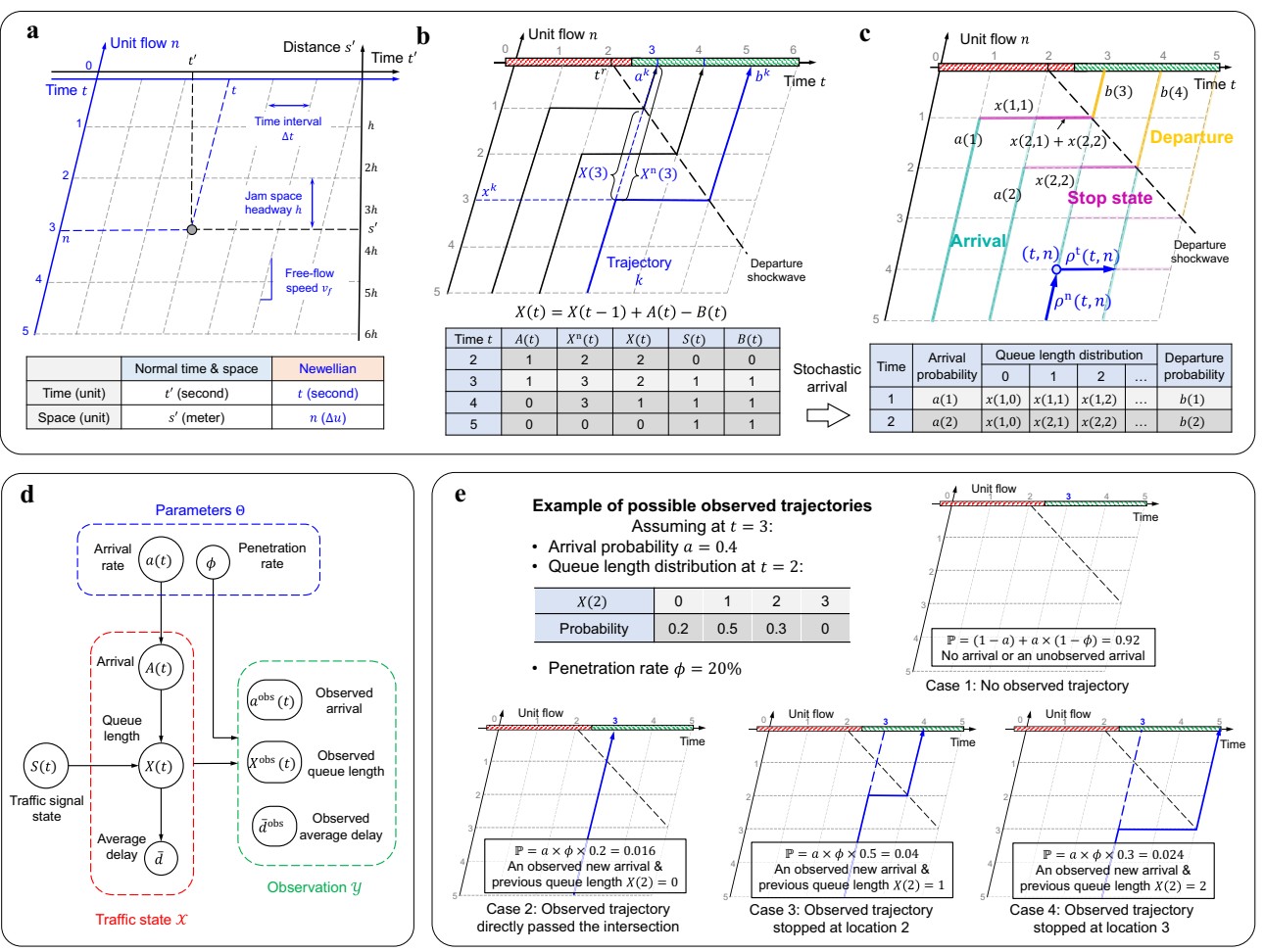

**Fig. 2 | Point-queue model under Newellian coordinates and PTS diagram.**
**a** Newellian coordinates $(t,n)$ and normal time-space coordinates $(t',s')$. The distorted grid (dashed lines) illustrates Newellian coordinates which are parameterized by free-flow speed $v_f$, time interval $\triangle t$, and jam space headway $h$.
**b** Point-queue representation under Newellian coordinates. Vehicle trajectories traveling on the edge of the distorted grid can be projected to a point-queue representation including arrival $A(t)$, departure $B(t)$, and queue length. There are two different queue lengths: the point queue $X(t)$ refers to the number of stopped vehicles while the spatial queue $X^n(t)$ refers to the location of the last stopped vehicle. $X(t)$ equals $X^n(t)$ minus the elapsed green time: by taking $t=3$ as an example, $X(3)=2$ and $X^n(3)=3$, their difference is $t-t^r=1$, where $t^r=2$ is the end of the red time. $S(t)$ denotes the traffic signal state where 0 and 1 indicate red and

green lights, respectively. **c** Probabilistic time-space (PTS) diagram. The point-queue representation can be projected back to the spatial-temporal space through the PTS diagram. The PTS diagram shows the spatial-temporal distribution of vehicle trajectories by drawing each edge and setting its transparency as the probability that there is vehicle traveling on it. The table below shows the input stochastic point-queue representations at each time: $a(t)$ and $b(t)$ is the probability that there is an arrival or departure at time $t$, respectively. $x(t,n)$ is the probability that the queue length is $n$ at time $t$, i.e., $x(t,n)=\mathbb{P}(X(t)=n)$. **d** Probabilistic graphical model (Bayesian network) under the stochastic point-queue model. **e** Example of possible observed trajectories at a certain time step. The examples illustrate the probability of observing a new vehicle trajectory at different stop locations.

respectively. The probability that there is a vehicle traveling at each edge can be calculated given the point-queue representations including arrival, queue length, and departure (Fig. 2c). By drawing each edge and setting its transparency as the associated probability, the PTS diagram directly shows the spatial-temporal distribution of vehicle trajectories. More detailed derivation of the PTS is included in the Methods section; an illustration video is also provided in Supplementary Movie 3.

The proposed stochastic point-queue model and PTS diagram enable us to establish a probabilistic graphical model (a Bayesian network) that connects observations with unknown traffic states and parameters (Fig. 2d). There are three main components: 1) Parameters Θ include the penetration rate and arrival rate. It could also contain other pre-determined and calibrated parameters such as free-flow speed, jam density, and turning ratios (Supplementary Section 5). These parameters are assumed to be stationary within a certain time of day. 2) Traffic state $\mathscr{X}$ including arrivals, departures, and queue lengths. 3) Observation $\mathscr{Y}$ comes from the vehicle trajectory data. By

assuming that the observed vehicles are randomly distributed among all vehicles, the penetration rate can be regarded as the probability of a vehicle being observed. Figure 2e gives an example that shows different possible observed trajectories at a certain time step with an assumed traffic state and parameters. A related illustration is also available in Supplementary Movie 3. Based on this probabilistic model (Fig. 2d), different statistical estimation methods can be applied to estimate both unknown traffic states and parameters from sparsely observed vehicle trajectories.

## Traffic state and parameter estimation
In this paper, the method of moments estimator is used to estimate traffic parameters including both the penetration rate and arrival rate. The intuition of this estimator is to match the average delay from the model-estimated value with the measurement from the observed trajectories. With the estimated traffic parameters, the traffic state can be directly derived through the stochastic point-queue model and the PTS diagram.

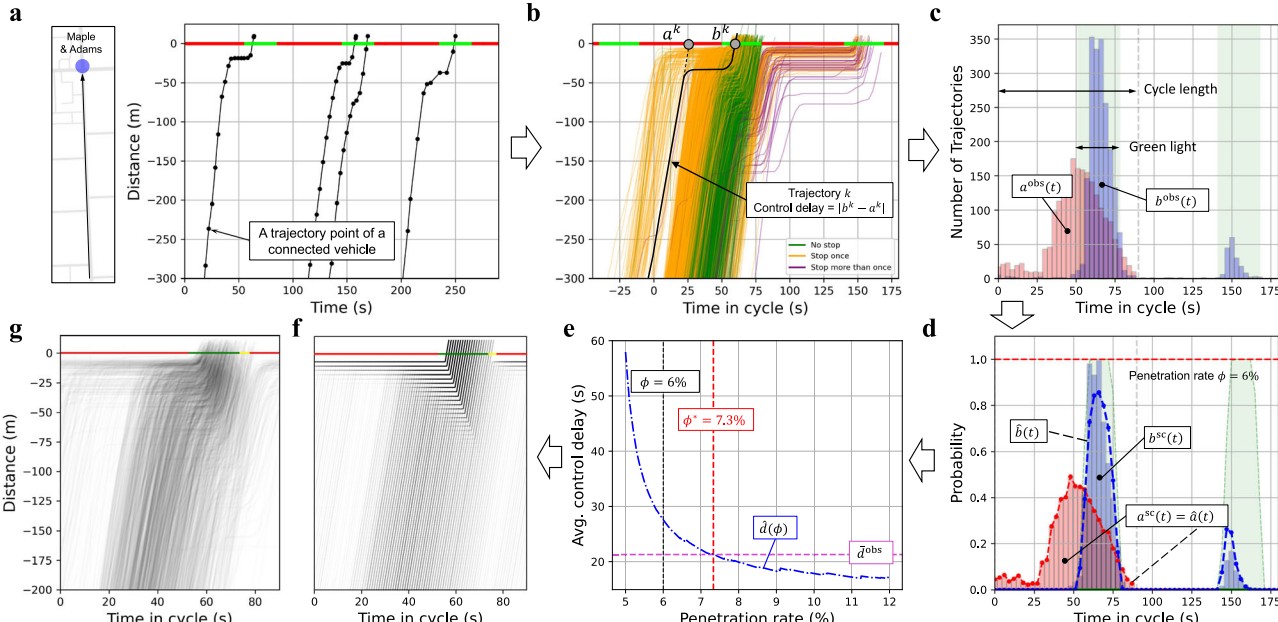

**Fig. 3 | Penetration rate estimation and reconstructed PTS diagram of an example movement.** All sub-figures are generated based on the mid-day period (10:00–15:00) of three consecutive weeks' (only weekdays from Monday to Friday) data from 03/07/2022 to 03/25/2022. **a** A short period (3 cycles) of the time-space (TS) diagram for a specific movement. **b** Aggregated TS diagram that aggregates trajectories of the same time of day (TOD) into one cycle. **c** Arrival and departure histograms ($a^{\text{obs}}(t), b^{\text{obs}}(t)$) generated from the aggregated TS diagram. All arrivals are within the first cycle (0–90 sec) while the departures might extend to the following cycle. **d** Arrival and departure profiles under a given penetration rate $\phi$ (6%

in the displayed case). Blue and red bars show the scaled arrival and departure profile, $a^{\text{sc}}(t)$ and $b^{\text{sc}}(t)$. The scaled arrival $a^{\text{sc}}(t)$ is used as the input arrival $\hat{a}(t)$, denoted by the red dashed line. The blue dashed line $\hat{b}(t)$ is the resulting departure profile derived from the queueing model. **e** Penetration rate estimation. The purple dashed line shows the average delay $\bar{d}^{\text{obs}}$ directly calculated from the observed trajectories while the dashed blue curve shows the model-estimated average delay $\hat{d}(\phi)$ under different penetration rates. The red vertical line shows the estimated penetration rate $\phi^*$ such that $\hat{d}(\phi^*) = \bar{d}^{\text{obs}}$. **f**, **g** Aggregated TS diagram (a complete cycle) and the reconstructed PTS diagram of the example movement.

Figure 3 is an illustration of parameter estimation for a specific movement (i.e., direction through an intersection). A related demonstration video is also available in Supplementary Movie 5. Figure 3a shows a short period (3 cycles) of the time-space (TS) diagram where the observed trajectories are sparse due to the low penetration rate. These trajectories of the same time of day (TOD) can be aggregated to one cycle to get the aggregated TS diagram as shown in Fig. 3b. Each trajectory is shifted by an integer number of cycles such that their arrival times are within the same cycle (Supplementary Movie 4). This aggregated TS diagram shows the average and recurrent traffic state of this movement. For each trajectory in Fig. 3b, the arrival and departure time in the Newellian coordinates can be extracted, and Fig. 3c shows the resulting arrival and departure time histograms. Note that since vehicle trajectories are aggregated according to their free-flow arrival times, some vehicles might depart in the following cycle if they fail to pass the intersection within the cycle in which they arrived.

Given sufficient vehicle trajectory data, the arrival and departure probability profiles can be estimated by scaling the histograms ($a^{\text{obs}}(t)$ and $b^{\text{obs}}(t)$ in Fig. 3c) according to the total number of cycles $N_c$, the unit flow per time step $\Delta u$, and a given penetration rate $\phi$:

$$a^{\text{sc}}(t) = \frac{a^{\text{obs}}(t)}{N_c \Delta u \cdot \phi} \quad b^{\text{sc}}(t) = \frac{b^{\text{obs}}(t)}{N_c \Delta u \cdot \phi}, \forall t \in \{1, 2, \cdots, T\} \tag{3}$$

where $T$ is the cycle length; $a^{\text{sc}}(t)$ and $b^{\text{sc}}(t)$ represent the scaled arrival and departure profiles (red and blue bars in Fig. 3d), respectively. This is also based on the assumption that the observed vehicles are randomly distributed among all vehicles.

The scaled arrival probability can be used as the input cyclic arrival for the stochastic point-queue model, that is, $\hat{a}(t + kT) = a^{\text{sc}}(t)$ for every cycle $k$ (red dashed line in Fig. 3d). The notation with "hat" means that it is a model-estimated value. Since both the traffic signal

state and input arrival are cyclic, the traffic state in a cycle will converge to a stationary distribution if the input arrival is strictly less than the capacity (Methods section). The stationary distribution of a traffic cycle is called the stationary traffic cycle. The blue dashed line in Fig. 3d is the resulting departure probability profile $\hat{b}(t)$ of the stationary traffic cycle; the average delay per vehicle $\hat{d}(\phi)$ can also be calculated. The dashed blue line in Fig. 3e shows how the model-estimated average delay $\hat{d}(\phi)$ changes with different given penetration rates $\phi$. When the penetration rate $\phi$ becomes higher, $\hat{d}(\phi)$ monotonically decreases since the input arrival also decreases according to Eq. (3). The optimal penetration rate $\phi^*$ can then be determined under which the model-estimated average delay $\hat{d}(\phi)$ matches the measurement $\bar{d}^{\text{obs}}$ from the observed trajectories, as illustrated by the red dashed line in Fig. 3e. At last, the arrival probability profile can be determined by applying the estimated penetration rate $\phi^*$ to Eq. (3). This completes the traffic parameter estimation of this movement.

By taking the estimated traffic parameter as the input, the traffic state can be directly derived based on the stochastic point-queue model. Figure 3f is the resulting PTS diagram of the stationary traffic cycle, which shows the spatial-temporal distribution of vehicle trajectories. Areas with darker colors indicate a higher probability that there are vehicles traveling on it. This PTS diagram directly corresponds to the aggregated TS diagram (Fig. 3g) since both diagrams show the average or recurrent traffic pattern in a cycle.

Similar estimation method can also be applied to a corridor consisting of multiple movements. Figure 4 shows the reconstruction of spatial-temporal traffic state of Adams Rd (northbound). Figure 4b is the corridor aggregated TS diagram, which is generated by combining the aggregated TS diagrams of all movements along the path. For visualization purposes, the aggregated TS diagrams for each movement are repeated over several cycles so that trajectories can traverse the whole corridor. Figure 4c shows the corresponding PTS diagram,

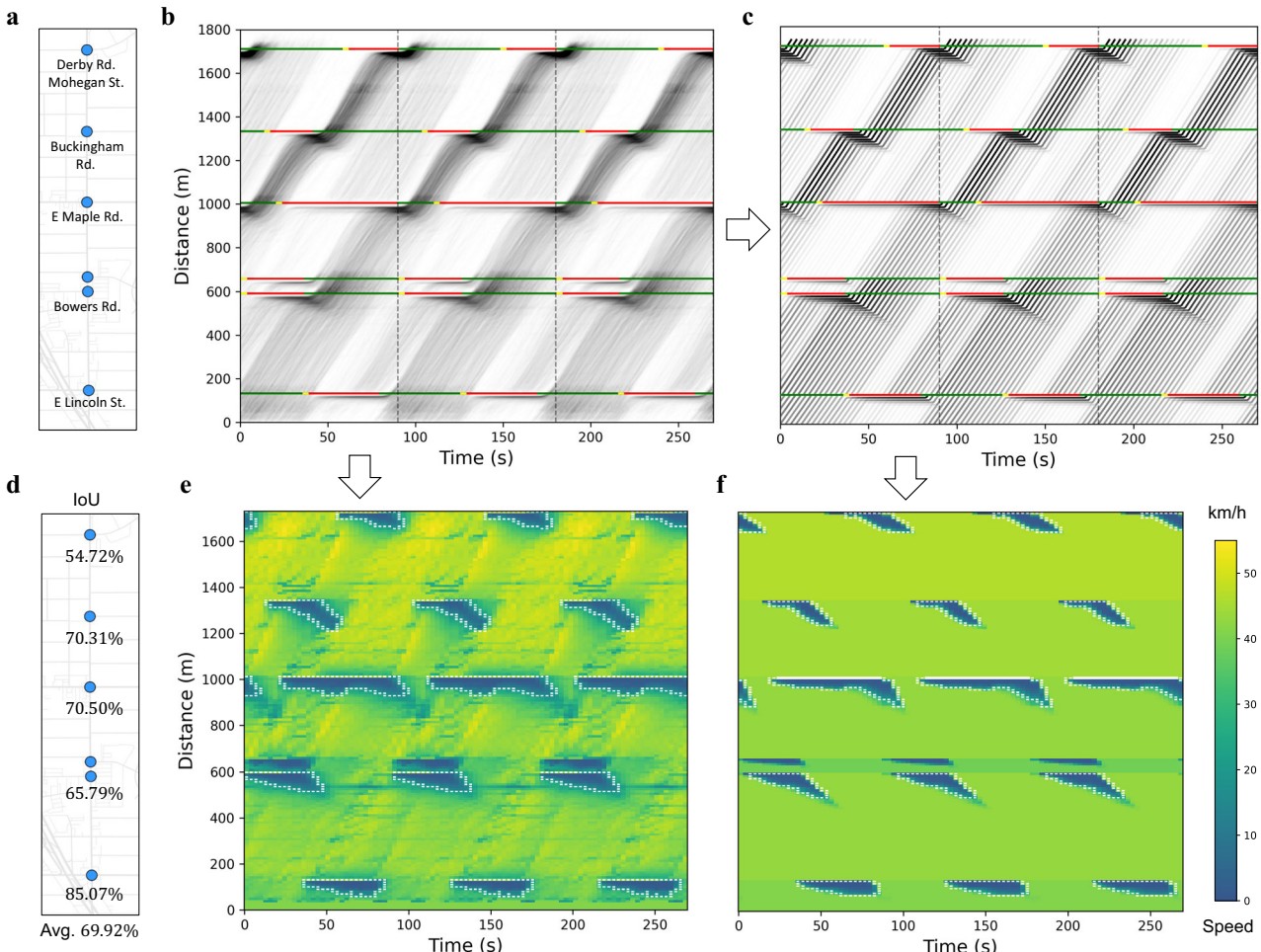

**Fig. 4 | Spatial-temporal traffic state reconstruction of Adams Rd (north-bound).** All sub-figures are generated based on the mid-day period (10:00–15:00) of three consecutive weeks' (only weekdays from Monday to Friday) data from 03/07/2022 to 03/25/2022. **a** Geometry of Adams Rd. **b** Aggregated TS diagram. **c** Reconstructed PTS diagram based on the calibrated stochastic traffic flow model. **e**, **f** Space-mean speed heatmaps that are generated based on the aggregated TS diagram and PTS diagram, respectively. The white dashed line is the boundary between the queueing area (dark color) and the free-flow area (light color), which are separated according to a pre-determined speed threshold. **d** IoU (Intersection over Union) of the queueing area for each intersection and average IoU of all intersections.

which are generated based on the calibrated stochastic traffic flow model.

Queueing area can be extracted from both TS diagrams (Fig. 4b, c) and used for verification purposes. Figure 4e, f are the space-mean speed heatmaps that are generated from the aggregated TS diagram and PTS diagram, respectively. To obtain the space-mean speed heatmaps, both TS diagrams are split into mesh grid according to certain spatial and temporal intervals (10 meters and 3 seconds), and the space-mean speed of each grid is then calculated as the total travel distance within the grid divided by the total travel time. The spatial-temporal space can be separated into queueing area and free-flow area according to a pre-determined speed threshold. White dashed lines are the boundary between queueing areas (dark color) and free-flow areas (light color). These boundary lines are also referred to as shockwaves in traffic flow theory[25,26], which separate the spatial-temporal traffic into different areas with relatively uniform traffic states. To validate the reconstructed traffic state as well as shockwaves, we use IoU (Intersection over Union) of the queueing area to quantify the similarity between ground truth and reconstructed heatmaps. The IoU of each signalized intersection is defined as the overlapped area between the ground truth and reconstructed queueing areas, divided by their combined area. Figure 4d reports the IoU of each intersection as well

as their average. An average IoU of around 70% indicates a good estimation.

## Diagnosis and optimization

The OSaaS traffic signal diagnosis module finds optimality gaps with respect to different signal timing parameters. Since the calibrated traffic flow model explicitly takes traffic signal parameters as an input, it can be directly used to predict network performance under different signal parameters by assuming unchanged traffic demand. The optimality gap can then be easily identified through either gradient-based or line search methods. For the signal timing parameters of isolated intersections such as cycle lengths and green splits, gradient-based methods are used since they usually do not require major changes. The sign of the gradient indicates the direction that could improve the system performance while the magnitude of the gradient quantifies the potential benefits. The output diagnostic results are categorized into different specific issues such as green split imbalances, insufficient cycle length, etc. These diagnostic results are directly used for generating new signal timing plans which essentially move a certain step size in the gradient direction.

Figure 5a–c is an illustration of traffic signal diagnosis for an isolated intersection. This isolated intersection utilizes a two-phase signal

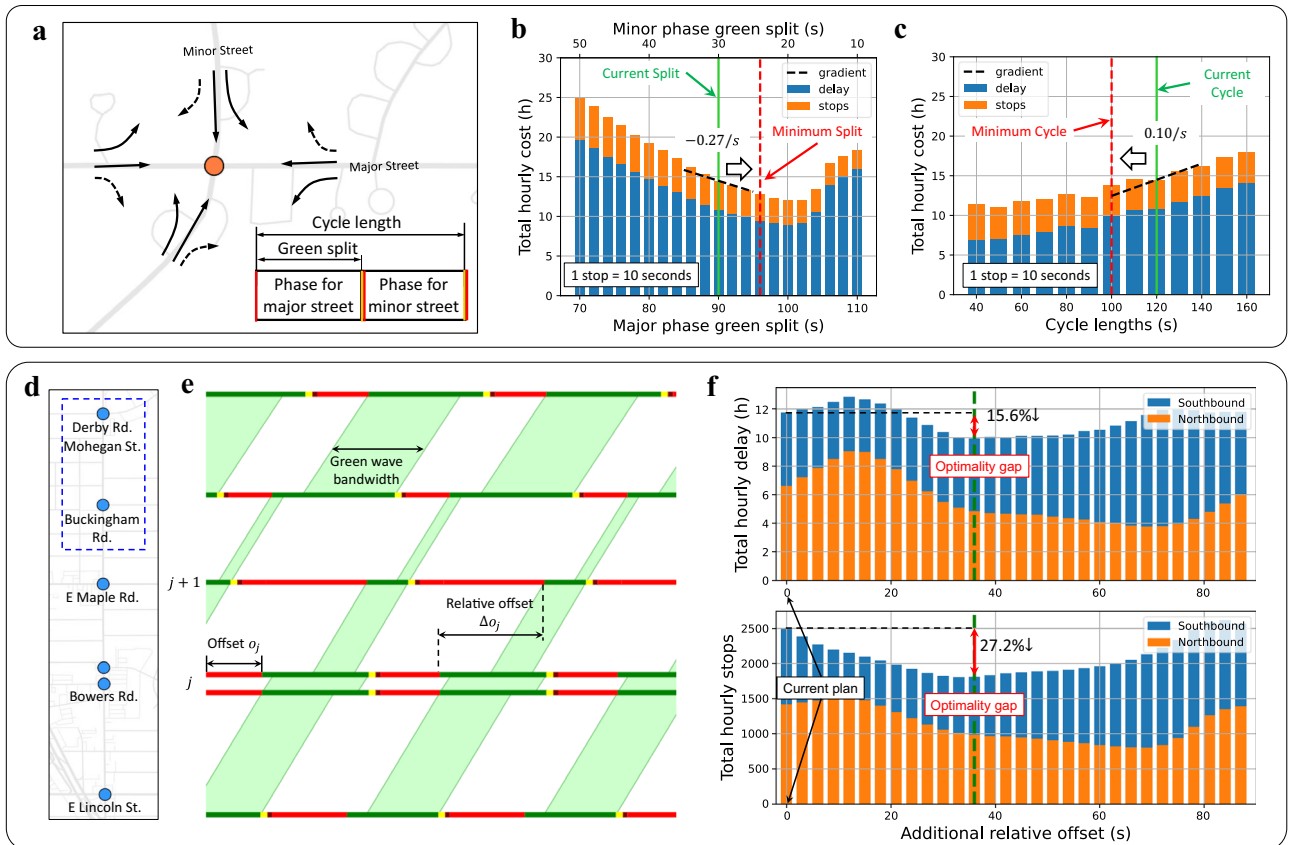

**Fig. 5 | Traffic signal diagnosis: isolated intersections & traffic signal coordination. a** Intersection geometry and traffic control parameters of an example isolated intersection: Quarton Rd. & Cranbrook Rd. **b** Green split diagnosis for the morning peak (AM, 07:00–10:00). The horizontal axis is the green duration for the major phase and vertical axis is hourly weighted summation of total delay and number of stops. **c** Cycle length diagnosis for the morning peak. This figure shows how the hourly total delay and number of stops change with different cycle lengths, where one stop is equivalent to 10 s of delay. **d** An example corridor: Adams Rd. which is composed of six signalized intersections. **e** Illustration of green band, offsets, and relative offsets. **f** Pair-wise coordination line search diagnosis for the midday period (MD, 10:00–15:00). Total delay & number of stops change with different additional relative offsets between the first two intersections.

operation where the major phase controls the major street, and the minor phase controls the minor street (Fig. 5a). Figure 5b, c are green split and cycle length diagnostic plots for the morning peak hours (AM). Figure 5b indicates that the total hourly cost of the intersection can be decreased by assigning more green time to the major phase while Fig. 5c indicates that this intersection can be improved by decreasing the cycle length. If the expected benefit exceeds a predetermined threshold, this intersection will be identified as one with the potential for improvement and the related parameters will be moved in the improving direction in the new signal timing plan.

We also propose a pair-wise coordination diagnosis method that efficiently detects better coordination opportunities as shown in Fig. 5d–f. Figure 5e demonstrates some basic traffic coordination concepts including green band, offsets, and relative offsets. The main objective of traffic coordination is to optimize the offsets of each intersection such that vehicles stop less when they traverse multiple intersections. For pair-wise coordination diagnosis, each pair of adjacent intersections are extracted as a sub-network. We then conduct line search on the relative offset between them to identify potential opportunities for better coordination. Taking the first two intersections as an example, Fig. 5f shows the predicted total delay and number of stops under different additional offsets. According to these curves, by adding an additional 36-second relative offset, the total delay and number of stops of these two intersections would decrease by about 16% and 27%, respectively.

To generate the new offsets along the entire corridor, we use a coordinate-descent program which aims at minimizing the total delay

and number stops. The details of the optimization program are provided in the Methods section. An illustration video is also available in Supplementary Movie 6. The proposed offset optimization method outperforms traditional green-band-based method[37,46,47] in two aspects: (1) it explicitly considers the vehicle distribution through the stochastic queueing model calibrated from vehicle trajectories; (2) it directly takes the total delay and number of stops as the objective function instead of green band which does not always indicate good coordination.

**Field implementation results**

The OSaaS system was tested in the City of Birmingham, Michigan, which has a total of 34 signalized intersections including three main corridors and some isolated intersections (Fig. 1a). More than three quarters of these intersections had not been retimed for more than 2 years. With the OSaaS system, two isolated intersections were detected with cycle/split issues and two of the three corridors were identified with coordination improvement opportunities. New signal timing plans were generated and implemented in late March 2022. Here we only show the results of the corridors while leaving the results of the isolated intersections for Supplementary Section 9. Please also refer to Supplementary Section 8 for a brief result of the performance evaluation of these intersections.

Tables 1–3 shows the implemented offset plans and the before-and-after comparison. For both corridors, offsets in three different TOD intervals were optimized including the morning peak hours (AM, 07:00–10:00), mid-day (MD, 10:00–15:00), and the evening peak

**Table 1 | Offset adjustment of Adams Rd**

| Side street | Time of day | Original offset (s) | New off-set (s) | Change (s) |
|---|---|---|---|---|
| Buckingham Ave. | 07:00–10:00 (AM) | 40 | 20 | −20 |
| | 10:00–15:00 (MD) | 40 | 20 | −20 |
| | 15:00–19:00 (PM) | 40 | 30 | −10 |
| Bower St. | 07:00–10:00 (AM) | 35 | 13 | −22 |
| | 10:00–15:00 (MD) | 35 | 13 | −22 |
| | 15:00–19:00 (PM) | 25 | 23 | −2 |
| Derby Rd./ Mohegan St. | 07:00–10:00 (AM) | 89 | 20 | −69 |
| | 10:00–15:00 (MD) | 89 | 21 | −68 |
| | 15:00–15:15 (PMa) | 89 | 31 | −58 |
| | 15:15–15:40 (PMb) | 89 | 31 | −58 |

The following table shows the new offsets and relative changes to the original offsets of Adams Rd.

**Table 2 | Offset adjustment of Old Woodward Ave**

| Side street | Time of day | Original off-set (s) | New off-set (s) | Change (s) |
|---|---|---|---|---|
| Merrill St. | 07:00–10:00 (AM) | 69 | 14 | −55 |
| | 10:00–15:00 (MD) | 52 | 22 | −30 |
| | 15:00–19:00 (PM) | 53 | 22 | −31 |
| Willits St. | 07:00–10:00 (AM) | 58 | 32 | −26 |
| | 15:00–19:00 (PM) | 77 | 39 | −38 |
| Brown St. | 07:00–10:00 (AM) | 39 | 22 | −17 |
| | 10:00–15:00 (MD) | 10 | 30 | 20 |
| | 15:00–19:00 (PM) | 15 | 30 | 15 |
| Oakland Ave. | 07:00–10:00 (AM) | 69 | 50 | −19 |

The following table shows the new offsets and relative changes to the original offsets of Old Woodward Ave.

**Table 3 | Before-and-after comparison of the offset optimization**

| Measurements | | Adams Rd. | | | | Old Woodward Ave. | | | |
|---|---|---|---|---|---|---|---|---|---|
| | | AM | MD | PM | Overall | AM | MD | PM | Overall |
| Average control delay (second) | Before | 13.92 | 11.88 | 16.03 | 13.89 | 19.03 | 17.72 | 18.58 | 18.29 |
| | After | 10.85 | 10.55 | 14.57 | 12.19 | 16.05 | 17.88 | 18.11 | 17.60 |
| | Change | −22.05% | −11.27% | −9.09% | −12.23% | −15.66% | 0.91% | −2.54% | −3.78% |
| Average number of stops | Before | 0.46 | 0.41 | 0.45 | 0.44 | 0.44 | 0.48 | 0.53 | 0.49 |
| | After | 0.33 | 0.33 | 0.41 | 0.36 | 0.42 | 0.44 | 0.45 | 0.44 |
| | Change | −28.69% | −21.13% | −10.55% | −18.51% | −6.09% | −8.5% | −14.63% | −10.77% |
| Space-mean speed (km/h) | Before | 36.42 | 38.94 | 34.30 | 36.51 | 17.97 | 17.76 | 17.34 | 17.64 |
| | After | 41.68 | 42.02 | 36.14 | 39.43 | 19.48 | 17.62 | 17.34 | 17.85 |
| | Change | 14.44% | 7.92% | 5.34% | 7.98% | 8.43% | -0.81% | -0.04% | 1.19% |

AM, MD, and PM represent morning peak, mid-day, and evening peak, respectively. Three different metrics are used including the average control delay, average number of stops, and space-mean speed for both directions of the corridor. Results in the table are based on three weeks' data (weekdays, from Monday to Friday) both before (03/07/2022-03/25/2022) and after (04/04/2022-04/22/2022) the implementation.

hours (PM, 15:00–19:00). Different metrics such as delay, number of stops, and space-mean speed were used to evaluate the performance of these two corridors. The average control delay and average number of stops of the corridor are calculated by dividing the total control delay and number of stops by the total number of trajectories, where a trajectory is defined as a vehicle passing one signalized intersection. Hence the delay and number of stops are reported per trajectory per intersection. The corridor's space-mean speed is calculated by dividing the total travel distance by the total travel time. Since only the offsets were changed and the green splits stayed the same, side street traffic was not influenced and is not included in the performance evaluation.

Table 3 shows the comparison of these three metrics before and after the offset optimization. Overall, the average control delay and the average number of stops of Adams Rd. decreased by 12% and 18%, respectively, while space-mean speeds increased by about 8%. For Old Woodward Ave., the average delay decreased by over 15% during the morning peak hours (AM) while the average number of stops decreased by over 14% during the evening peak hours (PM). However, for the mid-day period, the original offsets worked well and there was not a large optimality gap.

Figure 6 shows more details on how the new offsets fostered better traffic signal coordination along the corridors. Figure 6a–d shows the aggregated TS diagram of the Adams Rd. before and after the offset optimization. Each figure is generated from three con-secutive weeks of data collected at the mid-day TOD (10:00–15:00) during weekdays. As shown in Fig. 6c, d, the average delay and number of stops of the northbound through traffic decreased by over 18% and 40%; the southbound also outperformed the previous with a decrease

of 4% in both the average delay and number of stops. The dashed outlined areas M, N, K in Fig. 6a, b and the associated areas M', N', K' in Fig. 6c, d illustrate where the coordination improved. After the offset optimization, most of trajectories in these areas directly passed the downstream intersections without any stops. Blue lines are hypothe-tical trajectories that traverse the whole corridor which also have less delay and stops after the offset optimization.

## Discussion

This paper presents OSaaS, a large-scale traffic signal optimization system based on low penetration rate vehicle trajectory data. This system is cost-effective because it eliminates the manual signal retiming process and does not require installation and maintenance of road-side detectors. Without being restricted to installed locations, vehicle trajectory data is more scalable and is available for the whole road network, particularly for intersections with high traffic volumes. Besides, collective observation is more robust to equipment failure as it will not be affected if one vehicle loses its connectivity. As a closed-loop system, OSaaS continuously monitors urban traffic and can gen-erate new signal timing plans whenever sufficient historical data is accumulated. It significantly shortens each re-timing iteration, so a more responsive traffic signal retiming is feasible. Therefore, OSaaS provides a more scalable, sustainable, resilient, and efficient solution to the traffic signal re-timing practice, and could be applied to every fixed-time traffic signal in the world.

This study shows that, even at a low penetration rate, we can accumulate multi-day historical data to reconstruct the recurrent traffic state and use it for the periodical re-timing of fixed-time traffic

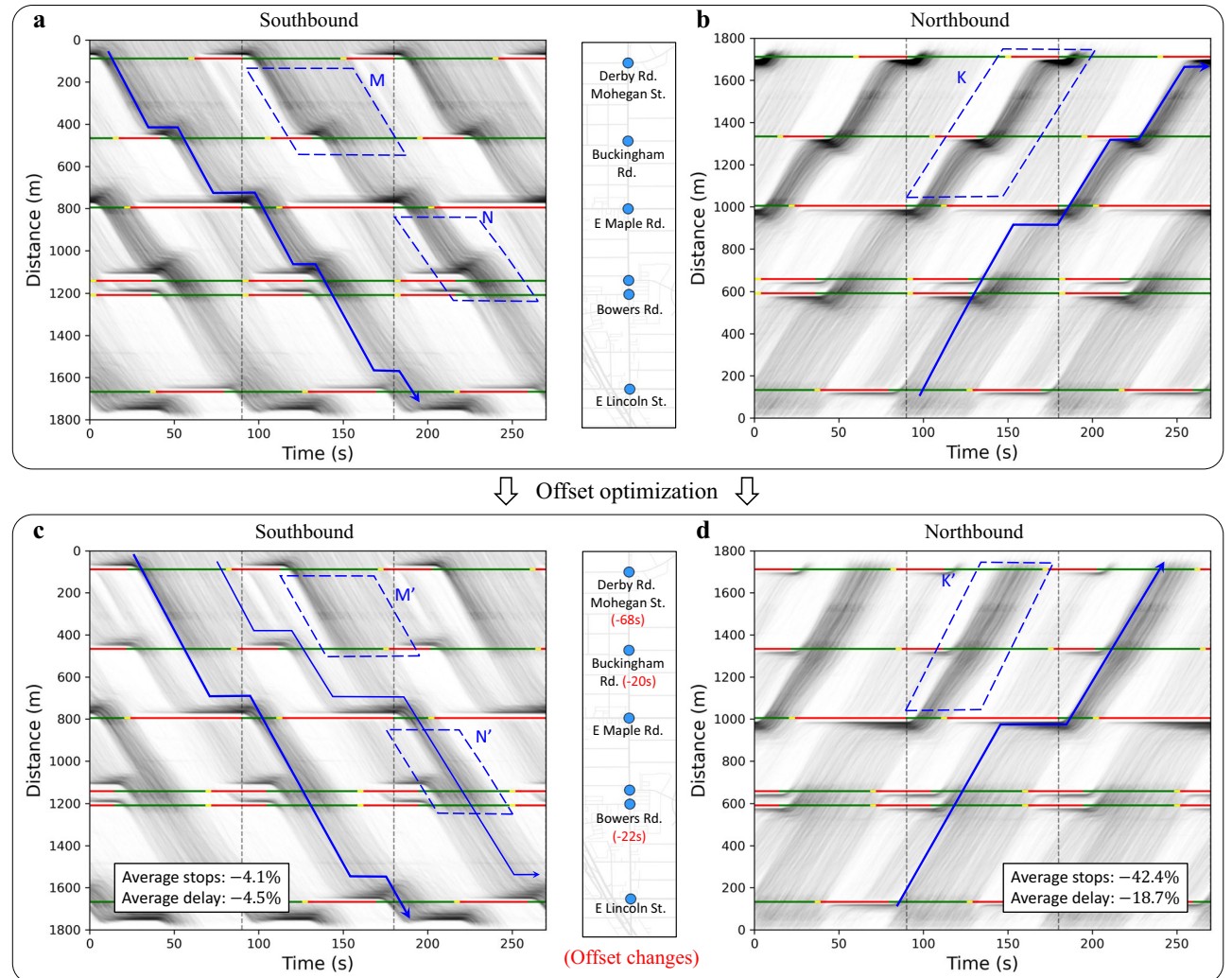

**Fig. 6 | Offset optimization example: mid-day period of Adams Rd. a–d** Show the southbound and northbound aggregated TS diagrams before and after the offset optimization. Before the offset optimization, the dashed outlined areas M, N, K in **a** and **b** show that the trajectories from the upstream queue arrived at the downstream during the red light. After the new offsets were implemented, the coordination in the same areas (M', N', and K' in **c** and **d**) were significantly improved: the upstream discharge queue directly passes the downstream intersection without stopping. The blue lines are hypothetical trajectories that traverse the whole corridor which also have less delay and number of stops.

signals. Such signal re-timing scheme can be further improved with real-time adjustments under certain conditions, e.g., high-volume congested intersections with a high risk of over-saturation or queue spillback. Both scenarios could be inferred from a small number of typical trajectories such as those fail to clear the intersection within one cycle or have a large queue length extending to the upstream intersection. However, the incomplete observation caused by low penetration rates may limit the accuracy of the real-time traffic state estimation. This will be improved in the future when more vehicles are connected.

In this paper, we performed a city-wide test of OSaaS in Birmingham, Michigan, and demonstrated that the delay and number of stops decreased for corridors and isolated intersections that were identified with optimality gaps. We believe that the proposed approach can be readily scaled to much-larger networks.

## Methods
### Discrete approximation, stochastic point-queue model, and PTS diagram
Each movement is modeled by a discrete stochastic point-queue model (i.e., a discrete queueing model) under the Newellian coordinates. For a certain movement, let $q^m$ and $z$ denote the saturation flow rate and the number of lanes, respectively. For each time interval $\Delta t$, the unit flow per time step $\Delta u$ at saturation flow rate is determined by: $\Delta u = q^m z \Delta t$. The discrete queueing model assumes binary arrival and departure, which means that both the arrival and departure are either 0 or $\Delta u$ for each time step. If the time interval is set properly, each unit arrival/departure could represent exactly one vehicle. For example, if a movement has two lanes, $z = 2$, and saturation flow rate $q^m = 1800$ veh/(lane · hour), then a unit arrival $\Delta u$ will be one vehicle if $\Delta t = 1$ sec. Let $h_0$ be the jam space headway with unit meter/(veh · lane), which is assumed to be a known constant. Then the jam space headway $h$ per unit flow (unit: meter/$\Delta u$) is given by:

$$h = \frac{\Delta u \cdot h_0}{z} = q^m h_0 \Delta t \tag{4}$$

Without loss of generality, we use $\Delta t = 1$ to simplify the notation, which means that time $t$ directly represents the number of time steps. For each time step, the binary arrival $A(t)$ follows a Bernoulli distribution with arrival probability $a(t)$, that is, $A(t) \sim \text{Bernoulli}(a(t))$. For simplification, arrivals at different time steps are assumed to be

independent. Let $B(t)$ and $X(t)$ represent the departure and queue length at time $t$, the queue length is updated by:

$$X(t) = X(t-1) + A(t) - B(t) = X'(t) - B(t) \qquad (5)$$

where $X'(t)$ is the intermediate queue length after the new arrival at time $t$. In each time step, the arrival happens before the departure such that vehicles can directly pass the intersection without stopping. Otherwise, every vehicle in the model would need to wait at least one time step before passing the intersection. The departure $B(t)$ is also binary and controlled by the traffic signal state $S(t)$:

$$\mathbb{P}(B(t)=1) \equiv b(t) = \mathbb{P}(X'(t) \geq 1) \cdot S(t) \qquad (6)$$

where $S(t)=0$ and $S(t)=1$ correspond to red and green lights, respectively. Equation (6) means that the departure will happen whenever the queue is not empty, and the traffic signal state is green. Let $x(t,k)$ be the pmf (probability mass function) of the queue length, which is the probability that the queue length $X(t)$ is $k$ at time $t$. Given an input arrival profile $a(t)$, the queue length distribution and departure can be updated recursively according to the following equations:

$$x'(t, k+1) = x(t-1, k) \cdot a(t) + x(t-1, k+1) \cdot (1 - a(t)) \qquad (7a)$$

$$x(t, k) = x'(t, k+1) \cdot S(t) + x'(t, k) \cdot (1 - S(t)), \forall k \geq 1 \qquad (7b)$$

$$x(t, 0) = x'(t, 1) \cdot S(t) + x'(t, 0) \qquad (7c)$$

$$b(t) = \sum_{k=1}^{\infty} x'(t, k) \cdot S(t) \qquad (7d)$$

The point queue $X(t)$ can be converted to the spatial queue $X^{\mathrm{n}}(t)$ through the following mapping function:

$$X^{\mathrm{n}}(t) = \Psi_{t,t'}(X(t)), \text{ where } \Psi_{t,t'}(k) = \begin{cases} k + (t - t^r)^+, & n > 0 \\ 0, & n = 0 \end{cases} \qquad (8)$$

where $t$ is the current time and $t^r$ is the end of the most recent red light. $(t - t^r)^+ \equiv \max\{0, t - t^r\}$ represents the elapsed green time. According to Eq. (8), for a traffic cycle starting from the red light, the point queue $X(t)$ is equivalent to the spatial queue $X^{\mathrm{n}}(t)$ during the red light ($0 \leq t \leq t^r$) while their difference is the elapsed green time $t - t^r$ during the green time ($t > t^r$). This deterministic mapping function is a result of the assumed deterministic driving behavior as well as the deterministic departure process.

Figure 2c shows the PTS diagram that can be projected from the discrete queueing model. Vehicle trajectories can only travel along the edges of the grid. As shown in Fig. 2c, $\rho^{\mathrm{n}}(t,n)$ and $\rho^{\mathrm{t}}(t,n)$ denote the probability that the vehicle travels on each vertical or horizontal edge, respectively. Edges in the grid can be divided into three categories including the arrival, departure, and stop states. For the stop state, the probability of each edge can be calculated by:

$$\rho^{\mathrm{t}}(t, \Psi_{t,t'}(n)) = \mathbb{P}(X(t) \geq n) = \sum_{k=n}^{\infty} x(t, k) \qquad (9)$$

where $\rho^{\mathrm{t}}(t, \Psi_{t,t'}(n))$ is the probability that there is a vehicle waiting from time $t$ to $t+1$ at point queue $X(t)=n$. The mapping function $\Psi_{t,t'}(\cdot)$ is used to transform the point queue to the corresponding spatial queue: $X^{\mathrm{n}}(t) = \Psi_{t,t'}(X(t)) = \Psi_{t,t'}(n)$. The probability in Eq. (9) is determined by the total probability that $X(t) \geq n$ since there will be a vehicle stopping at $X(t)=n$ as long as the point queue $X(t)$ is equivalent or larger than $n$.

For the departure edges as shown in Fig. 2c, the probability is calculated by:

$$\rho^{\mathrm{n}}(t, 0 : \Psi_{t,t'}(-1)) = \mathbb{P}(B(t)=1) = b(t) \qquad (10)$$

where $\rho^{\mathrm{n}}(t, 0 : \Psi_{t,t'}(-1))$ represents all the departure edges at time $t$ starting from the departure shockwave until leaving the intersection as shown in Fig. 2c.

For the arrival edges, the probability is calculated by:

$$\rho^{\mathrm{n}}(t, \Psi_{t,t'}(n)) = \mathbb{P}(A(t)=1) \cdot \mathbb{P}(X(t) < n) = a(t) \cdot \sum_{k=0}^{n-1} x(t, k), n \geq 1 \qquad (11)$$

$\rho^{\mathrm{n}}(t, \Psi_{t,t'}(n))$ represents the probability a vehicle traveling from $X(t)=n+1$ to $X(t)=n$. This event happens when there is a new arrival $A(t)=1$ and the queue length $X(t)$ is less than $n$.

Equations (9)–(11) show how the probability of a vehicle trajectory traveling on each edge in Fig. 2c is calculated from the discrete queueing model given by Eqs. (5)–(7). The probability of each edge is used as the edge's transparency in the diagram. In this way, the discrete queueing model is mapped to the probabilistic time-space (PTS) diagram and directly shows the spatial-temporal distribution of the vehicle trajectories. Supplementary Section 3 provides the derivation of the queueing model and the associated PTS diagram with a residual queue at the end of a cycle. Supplementary Section 4 introduces additional details related to effective green time calculation for both protected and permissive movements, as well as approximation of a network of movements.

## Traffic state and parameter estimation

Based on the probabilistic model given by Fig. 2d, the traffic estimation problem can be decomposed into two problems: 1) stationary parameter estimation and 2) traffic state estimation. Traffic parameters are estimated first since they provide prior information for the overall traffic state. By assuming that the penetration rate and arrival rate are stationary within a certain time of day, historical data can be aggregated to estimate these parameters. Different frequentist methods can be used. This paper uses the method of moments (MM) estimator. The intuition of the estimator is to find the parameters such that the observed average delay and the model estimated delay are equivalent:

$$\hat{d}\left(\hat{\Theta}_{\mathrm{MM}}\right) = \bar{d}^{\mathrm{obs}}, \qquad (12)$$

where $\hat{d}(\Theta)$ is the estimated average delay given parameter $\Theta$ while $\bar{d}^{\mathrm{obs}}$ is the average control delay directly measured from the observed trajectories.

Historical data from multiple cycles is needed for the method of moments estimator. Let $a^{\mathrm{obs}}(t)$ represent the total number of observed arrivals by aggregating trajectories from $N_c$ cycles (arrival histogram in Fig. 3c). Given the penetration $\phi$, the arrival rate of each time in the cycle can be estimated as:

$$\hat{a}(t) = \frac{a^{\mathrm{obs}}(t)}{N_c \Delta u \cdot \phi}, \forall t \in \{1, 2, \cdots, T\}. \qquad (13)$$

The queueing model can be configured with the estimated arrival profile and input traffic signal state. For fixed-time traffic signals, since both the traffic signal state and input arrival rate are cyclic with cycle $T$, that is, $S(t + kT) = S(t)$ and $a(t + kT) = a(t)$ for any cycle $k$, both the resulting departures and queue lengths will converge to a stationary distribution if the average traffic demand is within the traffic signal capacity:

$$\lim_{k \to \infty} X(t + kT) \to \bar{X}(t), \lim_{k \to \infty} B(t + kT) \to \bar{B}(t), \forall t \in \{1, 2, \cdots, T\}, \qquad (14)$$

where $\bar{X}(1:T)$ and $\bar{B}(1:T)$ represent the queue length and departure in a stationary traffic cycle, which can be calculated iteratively over cycles according to Eq. (7) (Supplementary Section 6).

Equation (14) also requires that the movement is strictly under-saturated on average: $\sum_{t=1}^{T} a(t) < \sum_{t=1}^{T} S(t)$. In this paper, we mainly focus on the fixed-time traffic signal optimization, which assumes stationary traffic state given a certain time of day (TOD). Therefore, we assume each movement to be under-saturated on average such that the stationary distribution given by Eq. (14) exists. This does not necessarily mean that the movement needs to be undersaturated for each individual cycle. Since the arrival process is stochastic, even if the arrival rate is strictly less than the capacity by average, the vehicle arrival could still be larger than capacity for some cycles and there will be a residual queue at the end of the cycle in this case. Please refer to Supplementary Section 3 for more details about the queueing model as well as the PTS diagram when the residual queue exists at the end of the cycle.

With the stationary arrival $\bar{A}(t)$ and queue length $\bar{X}(t)$, the average delay can be calculated according to Little's law[48]:

$$\bar{d} = \frac{\sum_{t=1}^{T} \mathbb{E}\left[\bar{X}(t)\right]}{\sum_{t=1}^{T} \mathbb{E}\left[\bar{A}(t)\right]} \tag{15}$$

By taking the estimated arrival given by Eq. (13) as the input, the model-estimated average delay will be a function of penetration rate $\phi$ and can be written as $\hat{d}(\phi)$. Then the penetration rate can be estimated according to the following formulation:

$$\phi^* = \arg\min_{\phi} \left[\hat{d}(\phi) - \bar{d}^{\text{obs}}\right]^2. \tag{16}$$

We also apply this method to estimate the penetration rates of multiple movements in a network of signalized intersections. For a movement with upstream arrival, the arrival from the upstream movement is estimated by an affine transformation of the upstream departure through a shift and scaling down (Supplementary Section 4.3). The shift duration is determined by the free-flow travel time and the relative offset, while the scaling coefficient is the turning ratio which can be directly calculated from the observed vehicle trajectory data. Since the penetration rates of different movements are close but different, the following centralized formulation is used to estimate the penetration rates of multiple movements in a network ($\mathcal{M}$ is the set of movements):

$$\boldsymbol{\phi}^* = \arg\min_{\boldsymbol{\phi}} \sum_{i \in \mathcal{M}} n_i^{\text{obs}} \left[\hat{d}_i(\phi_i) - \bar{d}_i^{\text{obs}}\right]^2 + \beta \mathbb{V}(\boldsymbol{\phi}), \tag{17}$$

where $\boldsymbol{\phi}$ is a column vector consisting of penetration rates of all the movements, $n_i^{\text{obs}}$ is the total number of observed trajectories of movement $i$. We slightly abuse notation $n$, which originally refers to the unit traffic flow in the Newellian coordinates, here represents the number of observed trajectories with superscript obs. $\mathbb{V}(\boldsymbol{\phi})$ is dispersion of the penetration rates of the individual movements weighted by total delay $n_i^{\text{obs}} \bar{d}_i^{\text{obs}}$:

$$\begin{aligned} \mathbb{V}(\boldsymbol{\phi}) &= \frac{1}{\sum_i n_i^{\text{obs}} \bar{d}_i^{\text{obs}}} \sum_{i \in \mathcal{M}} n_i^{\text{obs}} \bar{d}_i^{\text{obs}} \cdot (\phi_i - \bar{\phi})^2 \text{ where } \bar{\phi} \\ &= \frac{1}{\sum_i n_i^{\text{obs}} \bar{d}_i^{\text{obs}}} \sum_{i \in \mathcal{M}} n_i^{\text{obs}} \bar{d}_i^{\text{obs}} \phi_i. \end{aligned} \tag{18}$$

The first term of Eq. (17) is the summation of the delay difference between the traffic model and the observed trajectories weighted by the number of vehicles $n_i^{\text{obs}}$. The second term is a regularization through the dispersion of penetration rates weighted by the total delay

of each movement. $\beta$ is the coefficient of the regularization term. A larger $\beta$ will lead to more densely distributed penetration rates. If $\beta$ is sufficiently large, each movement will have the same penetration rate. Based on this centralized formulation, more congested movements with more delay will have a larger influence on the overall estimation program and will improve the estimation accuracy of the less congested movements.

## Traffic signal optimization

The calibrated traffic flow model can evaluate network performance under different traffic signal parameters. Let $s$ represent the traffic signal parameters including the cycle length, green splits, and offsets for all the signalized intersections in the network. $D(s)$ and $L(s)$ represent the total delay and number of stops derived from the queueing model. The traffic signal optimization problem can be formulated as:

$$s^* = \arg\min_{s \in \mathcal{S}} I(s), \text{ where } I(s) = D(s) + \alpha \cdot L(s), \tag{19}$$

where $I(\cdot)$ represents the overall performance index (PI), defined as the linear combination of the total delay and number of stops weighted by $\alpha$[49]. $\mathcal{S}$ denotes the feasible set of traffic signal parameters. Please note that, although we use the total delay and number of stops as the PI to be minimized, the choice of PI could be different and dependent on the needs of related stakeholders or traffic agencies. For example, it can be changed accordingly if a certain movement needs higher priority or the fairness among movements needs to be considered.

Different optimization programs are developed for both isolated intersections and corridor offset optimization. For isolated intersections, the re-timing of the cycle length and green splits is essentially a gradient-descent algorithm. For each signal re-timing iteration, new data is collected, and gradients are estimated from the calibrated traffic flow model. The new cycle length and green splits will be based on the original timing plan and move along the derivative direction for a certain step size (Supplementary Section 7.2).

Intersection offsets do not have much influence on capacity but could lead to better coordination with other intersections. For a specific TOD, the offset optimization for a corridor with $N$ signalized intersections can be formulated as:

$$\Delta\boldsymbol{o}^* = \arg\min_{\Delta o = [\Delta o_1, \cdots, \Delta o_{N-1}]} I(\Delta o_1, \Delta o_2, \ldots, \Delta o_{N-1}) \tag{20}$$

where the overall performance index $I(\cdot)$ is determined by the relative offsets $\Delta o = [\Delta o_1, \Delta o_2, \cdots, \Delta o_{N-1}]$. $\Delta o_j$ is the relative offset between intersection $j$ and $j+1$ as illustrated by Fig. 5e. Given the relative offsets $\Delta \boldsymbol{o}$, the offset $o_j$ of intersection $j$ is:

$$o_j = \left(\sum_{k=1}^{j-1} \Delta o_k\right) \bmod T \tag{21}$$

where $T$ is the cycle length. The optimization problem given by Eq. (20) can be solved by a coordinate-descent algorithm. For each iteration $i$, relative offsets are optimized sequentially according to:

$$\begin{aligned} \Delta o_j^i = \arg\min_{\Delta o_j} I\left(\Delta o_1^i, \cdots, \Delta o_{j-1}^i, \Delta o_j, \Delta o_{j+1}^{i-1}, \cdots, \Delta o_{N-1}^{i-1}\right), \\ \forall j \in \{1, 2, \cdots, N-1\} \end{aligned} \tag{22}$$

which can be solved through a line search program. $\Delta o_j^i$ denotes the relative offset of $j$th intersection in $i$th iteration. This iterative program will stop when the improvement in the latest iteration is less than a certain threshold.

## Vehicle trajectories, map, and signal timing data

The vehicle trajectory data in this work is from General Motors (GM) vehicles, which are equipped with GNSS (Global Navigation Satellite System) receivers and inertial measurement units (IMUs) that provide accurate vehicle position and dynamics information. These vehicles also have wireless communication capability (5G, LTE etc.) and support quick communication with cloud services. As a result, the vehicles can act as real-time mobile sensors that enable smart traffic signal operations. Trajectory point attributes include a unique trip ID, GNSS coordinates (latitude and longitude), timestamp, and speed. Their accuracy is roughly within 3–5 m, and they are received at a time interval of approximately 3 seconds. For the studied area (City of Birmingham, Michigan, US), there are approximately 2 million points and 25 thousand unique trips each day. The penetration rate is estimated to be around 7% according to this study.

The road network in this study is re-organized from OpenStreetMap[50], which is open-source and available online. Trajectories are matched to the road network so that we can convert raw GNSS coordinates to distance information of certain road segments[11,51–53]. Existing signal phase and timing (SPaT) data is extracted from signal work orders provided by Road Commission for Oakland County (RCOC). Offline vehicle trajectory data from 03/07/2022 to 03/25/2022 (three consecutive weeks) was used for modeling, diagnosis, and optimization. New signal timing plans were manually implemented by the RCOC on 03/31/2022 and 04/01/2022. After implementation, trajectory data collected from 04/04/2022 and 04/22/2022 (three consecutive weeks) was used for evaluation and comparison to the previous signal timing plans.

## Reporting summary

Further information on research design is available in the Nature Portfolio Reporting Summary linked to this article.

## Data availability

Raw map data used in this paper is extracted from the OpenStreetMap[50] and can be found at: https://www.openstreetmap.org. The raw vehicle trajectory data and SPaT data are not available due to data privacy laws. Processed data that support the findings of this study are publicly available at: https://doi.org/10.5281/zenodo.10493794[51]. Source data for figures are provided with this paper. Source data are provided with this paper.

## Code availability

The source code used to analyze experiment results and generate figures is publicly available at: https://doi.org/10.5281/zenodo.10493794[51].

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

## Acknowledgements

This research was partially funded by the U.S. Department of Transportation (USDOT) Region 5 University Transportation Center: Center for Connected and Automated Transportation (CCAT) of the University of Michigan (69A3551747105) and General Motors Holdings LLC (GAC3492). Any opinions, findings, and conclusions or recommendations expressed in this material are those of the authors and do not necessarily reflect the official policy or position of the Department of Transportation or the U.S. government.

## Author contributions

H.L., F.B. and A.J. conceived and led the research project. X.W., Z.J., S.S. and H.L. developed the OSaaS system concepts. X.W. and H.L. developed Newellian coordinates, stochastic point-queue model, PTS diagram, and related traffic state estimation methods. X.W., Z.J. and H.L. wrote the paper. X.W., Z.J. and Z.W. developed the algorithms for the stochastic point-queue model, PTS diagram, traffic parameter optimization, and traffic signal optimization. X.W., Z.J., Z.W. and C.Z. developed the algorithms to process the raw data including map, SPaT, and vehicle trajectories. V.K., F.B. and P. K. led the vehicle trajectory data collection. A.J., D.D. R.J. and G.P. provided the SPaT data of the tested intersections and conducted the field implementation. All authors provided feedback during the manuscript revision and results discussion. H.L. approved the submission and accepted responsibility for the overall integrity of the paper.

## Competing interests

H.L, X.W., Z.J., V.K., F.B., Z.W. and S.S. have filed a provisional patent application 18/308,996. The other authors declare no competing interests.
