## [Peer Review File · Nature Communications]

REVIEWER COMMENTS

Reviewer #1 (Remarks to the Author):

The study presents as traffic signal timing optimization methodology using connected vehicle data. The study is interesting, and timely. The application of connected vehicle data is novel. Following are my comments on the work:

- The mathematical formulations should be explained clearly. The paper is difficult to read and understand.
- It looks like a fixed penetration rate was used. However, the penetration rate of connected vehicle is dynamic and would vary for each cycle. How this aspect is handled in the present study?.
- How the shockwave properties such as shockwave area and shockwave speed validated?
- It is not clear how the signal timing, offset and signal phasing were optimized. Which method was used? Details regarding this should be added in the revised version.
- What is probability time-space diagram (PTS). The concept should be explained in brief. How the PTS varies compared to the TS diagram.
- It is mentioned that stochasticity is introduced in the model, but how it is introduced and handled is not clear at all.
- The details regarding the saturation level of the intersections is missing. Whether the selected intersections are under-saturated, saturated or over-saturated?
- How the methodology would perform for oversaturated intersections? A brief discussion on this should be added.
- Authors considered performance measures such as control delay, number of stop and space-mean speed. Why other performance measures such as queue length, intersection capacity, and discharge profile not considered. Multiple performance measures should be used for a holistic assessment of the methodology.

Reviewer #2 (Remarks to the Author):

The authors investigated a traffic light optimization problem using the data from connected vehicles. One of the contributions of the work is implement it in practice. Some issues could be addressed:

- The problem is well-investigated in the literature (DOIs: 10.1109/TITS.2015.2471812). Further comparison with related works are needed.
- Problem formulation is not provided.
- An issue of fairness (equilibrium) among vehicles is not considered.
- How new signal timing plans were implemented in field? OSaaS system is not described in details. Reproducibility is not guaranteed.
- Recently, deep reinforcement learning is emerging as a powerful tool for optimizing the traffic light (E.g., DOI: 10.1016/j.knosys.2022.108304). It is missing in the manuscript. Different optimization methods should be considered.
- Some works using computer vision to estimate the traffic volume and adjust the signals in real-time. Please compare the advantages and disadvantages.
- One of the research trend is signal-free intersection (E.g., DOIs: 10.1109/MITS.2017.2743167, 10.1109/ACCESS.2018.2871337). It is worth to discuss.

Reviewer #3 (Remarks to the Author):

The manuscript presents a new traffic signal timing system that uses connected vehicle data. They first connect a stochastic point-queue model with vehicle trajectories under a proposed Newellian (inspired by Newell's car-following behavior) coordinates to construct spatiotemporal traffic states. They then use an optimization algorithm to update traffic signal parameters. They implemented the signal optimization on a real-world network in Michigan and observed improvements in traffic delay and the number of stops using their proposed framework. Overall, the manuscript is very well-written, covers an intriguing topic, and presents a novel and valid methodology. My comments/suggestions are as below.

It is stated in the manuscript, for instance, that "we present the world's first large-scale cloud based traffic signal re-timing system that uses a small percentage of connected vehicle trajectories as the only input without reliance on any infrastructure-based detectors.", "By utilizing the trajectory data as the only input and not requiring any additional infrastructure, OSaaS provides a more scalable and economical solution to traffic signal retiming which can potentially be applied to every fixed-time traffic signal in the world.", or "We present the world's first large-scale cloud-based traffic signal optimization system (OSaaS: Optimizing Signals as a Service) based on vehicle trajectory data collected by connected vehicle service providers, which could be independent of traffic management agencies."

These sentences imply that this study is the first one that optimizes signal timing using vehicle trajectory data. However, there are many other studies in the literature that utilize only this data and not detectors'

data for signal optimization. In order to provide a comprehensive evaluation of the manuscript's contributions, I would recommend discussing its unique aspects in relation to these existing studies. Emphasizing the distinctive contributions on these points would enhance the manuscript's overall clarity and significance. For your reference, I have included two examples of studies that utilize vehicle trajectory data below:

Xu, Biao, Xuegang Jeff Ban, Yougang Bian, Wan Li, Jianqiang Wang, Shengbo Eben Li, and Keqiang Li. "Cooperative method of traffic signal optimization and speed control of connected vehicles at isolated intersections." *IEEE Transactions on Intelligent Transportation Systems* 20, no. 4 (2018): 1390-1403.

Li, Wan, and Xuegang Ban. "Connected vehicles based traffic signal timing optimization." *IEEE Transactions on Intelligent Transportation Systems* 20, no. 12 (2018): 4354-4366.

The manuscript utilized connected vehicle data from General Motors (GM) for the case study. The usage of the term 'connected vehicle' might be slightly confusing in this context since the GM vehicles are not connected to each other or an infrastructure. To the best of my knowledge, the GM data consists of GPS data from these vehicles, which can be considered partially connected. I would suggest clarifying the definition of connected vehicles and providing more specific details regarding the data.

Response to the Reviewers' Comments

The authors would like to thank the reviewers for their constructive comments. In response, we have carefully revised the paper and highlighted our changes in red. We believe the revised paper addresses all the questions raised by the reviewers. The following is a point-to-point response to all comments.

Reviewer #1 (Remarks to the Author)

The study presents a traffic signal timing optimization methodology using connected vehicle data. The study is interesting, and timely. The application of connected vehicle data is novel.

Following are my comments on the work:

- 1. The mathematical formulations should be explained clearly. The paper is difficult to read and understand. It looks like a fixed penetration rate was used. However, the penetration rate of connected vehicle is dynamic and would vary for each cycle. How is this aspect handled in the present study?*

Response: We have made a thorough revision to the paper, particularly the first two subsections in the Results section. We believe the mathematical formulation is clearer and easier to understand. A notation table has also been added to the Supplementary Section 1. The Results section only contains key parts of the formulation while the detailed mathematical descriptions are in the Methods section as well as Supplementary Materials. We also recommend the reviewer to look at the related Supplementary Movies.

Concerning the penetration rate, this paper focuses on optimizing fixed-time traffic signals, which are timed to control the average traffic state within certain time of day (TOD) periods. Therefore, we assume that the traffic parameters including the arrival rate and penetration rate are stationary for a movement within the same TOD. Besides, we also assume that connected vehicles are randomly distributed among all vehicles, so the penetration rate is essentially the probability that a vehicle is observed. With these assumptions, the actual realized penetration rate could be different for a specific cycle or a certain short period, but it will converge to the true penetration rate when the study period is long enough. Since we only need the average traffic state within a TOD for fixed-time traffic signal optimization, we do not care about the actual realized penetration rate for each specific cycle or a shorter period but only the average penetration rate (ϕ in the paper) over a sufficiently long period.

- 2. How are the shockwave properties such as shockwave area and shockwave speed validated?*

Response: We thank the reviewer for this inspiring question. One of the major contributions of this paper is reconstructing the spatial-temporal vehicle distribution (PTS diagram) from the simple stochastic point-queue model. Previously, we used the RMSD (root-mean-square derivation) of the traffic densities to compare the generated PTS diagram with the aggregated TS diagram. Inspired by the reviewer, we noticed that a better method for validation is to compare the shockwave area as well as the shockwave line (i.e., boundary between different shockwave areas). For traffic flow in urban traffic networks, for simplification, the traffic state can be roughly divided into stop state and free-flow state, so do the shockwave areas. Since the queueing area (i.e., stop state) is more of interest, we propose the following method for verification. Please refer to Line 261, Page 7 in the main paper:

“Queueing area can be extracted from both TS diagrams (Fig.4b-c) and used for verification purposes. Fig. 4e-f are the space-mean speed heatmaps that are generated from the aggregated TS diagram and PTS diagram, respectively. To obtain the space-mean speed heatmaps, both TS diagrams are split into mesh grid according to certain spatial and temporal intervals (10 meters and 3 seconds), and the space-mean speed of each grid is then calculated as the total travel distance within the grid divided by the total travel time. The spatial-temporal space can be separated into queueing area (red) and free-flow area (green) according to a pre-determined speed threshold. Blue dashed lines are the boundary between queueing area and free-flow area. We use IoU (Intersection over Union) of the queueing area to quantify the similarity between ground truth and reconstructed heatmaps. The IoU of each signalized intersection is defined as the overlapped area between the ground truth and reconstructed queueing areas, divided by their combined area. Fig. 4d reports the IoU of each intersection as well as their average. An average IoU of around 70% indicates a good estimation.

Fig. 4 Spatial-temporal traffic state reconstruction of Adams Rd (northbound). All sub-figures are generated based on the mid-day period (10:00-15:00) of three consecutive weeks’ (only weekdays from Monday to Friday) data from 03/07/2022 to 03/25/2022. (a) Geometry of Adams Rd. (b) Aggregated TS diagram. (c) PTS diagram of Adams Rd. (e-f) Space-mean speed heatmaps that are generated based on the aggregated TS diagram and PTS diagram, respectively. The dark blue dashed line is the boundary between the queueing area (red) and the free-flow area (green), which are separated according to a pre-determined speed threshold. (d) IoU (Intersection over Union) of the queueing area for each intersection and average IoU of all intersections.”

3. *It is not clear how the signal timing, offset and signal phasing were optimized. Which method was used? Details regarding this should be added to the revised version.*

Response: The optimization of signal timing for isolated intersections is available in the Supplementary Materials, Section 7.2. We have also moved the offset optimization for a corridor from the Supplementary Materials to the Methods section, it is written as (Line 518, Page 18 in the main paper):

“Intersection offsets do not have much influence on capacity but could lead to better coordination with other intersections. For a specific TOD, the offset optimization for a corridor with N signalized intersections can be formulated as:

$$\Delta \mathbf{o}^* = \underset{\Delta \mathbf{o} = [\Delta o_1, \dots, \Delta o_{N-1}]}{\operatorname{argmin}} I(\Delta o_1, \Delta o_2, \dots, \Delta o_{N-1}) \quad (20)$$

where the overall performance index $I(\cdot)$ is determined by the relative offsets $\Delta \mathbf{o} = [\Delta o_1, \Delta o_2, \dots, \Delta o_{N-1}]$. Δo_j is the relative offset between intersection j and $j + 1$ as illustrated by Fig. 4d. Given the relative offsets $\Delta \mathbf{o}$, the offset o_j of intersection j is:

$$o_j = \left(\sum_{k=1}^{j-1} \Delta o_k \right) \bmod T \quad (21)$$

where T is the cycle length. The optimization problem given by Equation (20) can be solved by a coordinate-descent algorithm. For each iteration i , relative offsets are optimized sequentially according to:

$$\Delta o_j^i = \underset{\Delta o_j}{\operatorname{argmin}} I(\Delta o_1^i, \dots, \Delta o_{j-1}^i, \Delta o_j, \Delta o_{j+1}^{i-1}, \dots, \Delta o_{N-1}^{i-1}), \forall j \in \{1, 2, \dots, N-1\} \quad (22)$$

which can be solved through a line search program. Δo_j^i denotes the relative offset of j th intersection in i th iteration. This iterative program will stop when the improvement in the latest iteration is less than a certain threshold.”

4. *What is probability time-space diagram (PTS). The concept should be explained in brief. How the PTS varies compared to the TS diagram.*

Response: The PTS diagram projects the stochastic point-queue model back to a spatial-temporal space to show the spatial-temporal vehicle trajectory distribution. To make it clearer, we have rewritten description as follows (Line 183, Page 5):

“Fig. 2b shows how the stochastic point-queue model can be projected back to the spatial-temporal space using the probabilistic time-space (PTS) diagram. As aforementioned, vehicles only travel on the edges of the Newellian coordinates. Let $\rho^n(t, n)$ and $\rho^t(t, n)$ denote the probability that there are vehicles travelling on the vertical and horizontal edges, corresponding to the free-flow and stop states, respectively. The probability that there is a vehicle travelling at each edge can be calculated given the point-queue representations including arrival, queue length, and departure (Fig.2b). By drawing each edge and setting its transparency as the associated probability, the PTS diagram directly shows the spatial-temporal distribution of vehicle trajectories. More detailed derivation of the PTS is included in the Methods section; an illustration video is also provided in Supplementary Movie 3.”

As a comparison, the (aggregated) TS diagram directly draws the real-world vehicle trajectories while the PTS diagram shows the spatial-temporal distribution of vehicle trajectories that is generated from the stochastic point queue model. In this paper, we show and compare the aggregated TS diagram and PTS diagram in main paper Fig. 4. The aggregated TS diagram (Fig. 4b) is obtained by drawing each observed vehicle trajectory. The PTS diagram (Fig. 4c) is obtained based on the calibrated stochastic point-queue model. Both PTS diagram and aggregated TS diagram show the average or recurrent traffic pattern (spatial-temporal vehicle distribution) in a cycle.

5. *It is mentioned that stochasticity is introduced in the model, but how it is introduced and handled is not clear at all.*

Response: There are multiple major sources of randomness for the partially observable system, which include: 1) stochastic traffic demand; 2) stochastic driving behaviors; 3) uncertainty caused by incomplete observation (low penetration rate). In this paper, we explicitly consider the first and the third factors while ignoring the second one. Here is the justification in the main paper (Line 141, Page 4):

“... the proposed Newellian coordinates, which are established on the assumption that all vehicles follow a homogeneous deterministic Newell’s car-following model. This assumption holds for the urban traffic featured by interrupted flow, where stop-and-go is the dominant characteristic of vehicle trajectories and vehicle delay mainly comes from the stopping time caused by traffic signals and queuing. When the penetration rate is low, most of the estimation uncertainty arises from incomplete observation and stochastic traffic demand. Therefore, we ignore heterogeneity and stochasticity of driving behaviors in this paper.”

With respect to the stochastic traffic demand, we assume that the arrival at each time interval follows a Bernoulli distribution. Please refer to Line 177, Page 5:

“It is assumed that the vehicle arrival $A(t)$ at each time follows a Bernoulli distribution with arrival rate $a(t)$. There is a departure, i.e., $B(t)=1$, whenever the traffic light is green, and the existing queue is not empty. In this way, we have specified the transition of the stochastic queueing model, and the queue length distribution can be derived given the input arrival and traffic signal state.”

To model the uncertainty caused by incomplete observation, we assume that each vehicle has a certain probability (i.e., the penetration rate) of being an observed connected vehicle.

6. *The details regarding the saturation level of the intersections are missing. Whether the selected intersections are under-saturated, saturated, or over-saturated?*

Response: Based on the available trajectory data, one metric that is directly related to the saturation level is the proportion of the split failure cycles. A cycle is oversaturated with split failure if a vehicle takes more than one signal cycle to pass the intersection. We have added the following table to Supplementary Materials (Supplementary Section 8).

Intersection	Average control delay (s)	Average number of stops	Split failure ratio (%)	Level of Service (A-F)
Adams Rd./Derby Rd.	16.46	0.490	2.80%	B
...

7. *How would the methodology perform for oversaturated intersections? A brief discussion on this should be added.*

Response: Please refer to the following discussion that has been added to the Methods section (Line 471, Page 17):

“In this paper, we mainly focus on the fixed-time traffic signal optimization, which assumes stationary traffic state given a certain time of day (TOD). Therefore, we assume each movement to be undersaturated on average such that the stationary distribution given by Equation (14) exists. This does not necessarily mean that the movement needs to be undersaturated for each individual cycle. Since the arrival process is stochastic, even if the arrival rate is strictly less than the capacity by average, the vehicle arrival could still be larger than capacity for some cycles and there will be a residual queue at the end of the cycle in this case. Please refer to Supplementary Section 3 for more details about the queuing model as well as the PTS diagram when the residual queue exists at the end of the cycle.”

If the movement has a large traffic volume which is close to the capacity, there will be a large probability that the queue is not cleared within a cycle. In this case, we will have a split failure for the cycle and a residual queue at the end of the cycle. In Supplementary Section 3, we have shown the details about the point-queue model as well as the associated PTS diagram for this case.

8. *Authors considered performance measures such as control delay, number of stop and space-mean speed. Why other performance measures such as queue length, intersection capacity, and discharge profile are not considered. Multiple performance measures should be used for a holistic assessment of the methodology.*

Response: In this paper, we use control delay, number of stops, and space mean speed as performance measures as they can be directly measured from the vehicle trajectory data. Therefore, we use both control delay and number of stops in the objective function for signal optimization. We did not include queue length and other performance measures because they are not included in the objective function for the signal optimization, although all of them can be easily estimated from the queuing model developed in the paper.

Reviewer #2 (Remarks to the Author)

The authors investigated a traffic light optimization problem using data from connected vehicles. One of the contributions of the work is implementing it in practice. Some issues could be addressed:

1. *The problem is well-investigated in the literature (DOIs: 10.1109/TITS.2015.2471812). Further comparison with related works is needed.*

Response: The survey paper mentioned by the reviewer focuses on cooperative intersection management, particularly with automated vehicles. Most related studies in the survey paper assume a high penetration rate of connected and automated vehicles (CAV). They usually assume a 100% CV (connected vehicles) or there is a road-side detector (including camera) that can observe complete information within the intersection. The penetration rate of AV (automated vehicles) also needs to be high enough to have significant improvement. In addition, these studies were only tested in simulation environments. In this paper, we aim at optimizing traffic signals using only connected vehicle trajectory data with low penetration rate ($\leq 10\%$). No automated vehicles are assumed. The traffic state of the intersection cannot be completely observed because of the low penetration rate. The developed algorithms in this paper were also implemented and tested in the real world, not in the

simulation environment. We have added the following paragraph to the Introduction section (Line 69, Page 2) to further clarify the focus of our study:

“This paper focuses on the optimization of fixed-time traffic signals using connected vehicle trajectories and does not rely upon any road-side detectors (e.g., loop detectors, cameras). A vehicle is considered to be a connected vehicle if its location is known such that its trajectory can be constructed. Although many existing studies have investigated traffic signal control with connected and automated vehicles (CAV)¹⁻⁷, they assume a high penetration rate (i.e., the proportion of CAVs to the overall number of vehicles), which is not realistic in the current practice. In this study, we aim at optimizing traffic signals utilizing vehicle trajectories at the currently available market penetration rate. In this case, one major challenge is the resulting sparse and incomplete observation of the overall traffic state.”

One of the major reasons that we are interested in only utilizing vehicle trajectory data is that this will give us a more scalable solution compared with detector-based solutions. Vehicle trajectory data is available at almost every intersection, not restricted to installed locations.

2. Problem formulation is not provided.

Response: As stated in in Line 74, *“in this paper, we aim at optimizing traffic signals utilizing vehicle trajectories at the currently available market penetration rate.”* To solve this problem, in the Results section, we propose a stochastic traffic flow model based on newly introduced Newellian coordinates (Results Subsection 1), which is applied to estimate the overall traffic state and parameters (Results Subsection 2). The calibrated traffic model is further utilized for traffic signal diagnosis and optimization (Results Subsection 3).

3. An issue of fairness (equilibrium) among vehicles is not considered.

Response: We will discuss the fairness among vehicles in two aspects:

- a. Fairness between observable connected vehicles and unobservable normal vehicles. Although we only utilize observed connected vehicle trajectories for traffic signal optimization, theoretically, there is no bias between observable connected vehicles and unobservable normal vehicles. Before performing the traffic signal optimization, the overall traffic volume is estimated based on the assumption that connected vehicles are randomly distributed among all vehicles. The estimation will be unbiased if this assumption holds. We should mention that this assumption is usually true, particularly for the arriving traffic in a specific movement: connected vehicles cannot determine when they arrive at a movement and thereby, they just randomly distribute among overall arriving vehicles. By taking the estimated overall traffic volume as the input, the generated fixed-time signal timing plan would not put more priority on connected vehicles over unobserved vehicles.
- b. Fairness between different movements. The current objective function used in the traffic signal optimization program is the weighted sum of total average delay and total number of stops. This is a commonly used objective function of traffic signal control, but it is true that this does not consider fairness among different movements. This can be improved by choosing different objective functions. We have added the following brief discussion to Line 509, Page 18:

“Please note that, although we use the total delay and number of stops as the PI to be minimized, the choice of PI could be different and dependent on the needs of related stakeholders or traffic agencies. For example, it can be changed accordingly if a certain movement needs higher priority or the fairness among movements needs to be considered.”

4. *How new signal timing plans were implemented in the field? OSaaS system is not described in detail. Reproducibility is not guaranteed.*

Response: Table 1 a-b in the main paper shows the change of the offsets for both corridors. Other than offsets, all the other signal timing parameters remained the same. New signal timing plans were implemented by the Road Commission for Oakland County (RCOC). They sent out a traffic engineer to each intersection and manually changed the parameters through the traffic signal controllers. The implementation is introduced in Line 545, Page 19:

“Existing signal phase and timing (SPaT) data is extracted from signal work orders provided by Road Commission for Oakland County (RCOC). Offline vehicle trajectory data from 03/07/2022 to 03/25/2022 (three consecutive weeks) was used for modeling, diagnosis, and optimization. New signal timing plans were manually implemented by RCOC on 03/31/2022 and 04/01/2022. After implementation, trajectory data collected from 04/04/2022 and 04/22/2022 (three consecutive weeks) was used for evaluation and comparison.”

The framework of OSaaS is provided in Fig. 1e. In the Results section, we have sequentially introduced 1) what model is used, 2) how we estimate unknown parameters, 3) how we perform the diagnosis, 4) results of the field implementation. We believe this content covers the complete OSaaS pipeline. While the Results section only contains key methods and results, more details are available in the Methods section and Supplementary Information.

5. *Recently, deep reinforcement learning is emerging as a powerful tool for optimizing the traffic light (E.g., DOI: 10.1016/j.knosys.2022.108304). It is missing in the manuscript. Different optimization methods should be considered.*

Response: In this paper, we focus more on modeling and system identification instead of the actual algorithms that are used to generate new signal timing plans (e.g., RL or other optimization-based methods). A well-calibrated model is an essential input for most RL and optimization-based methods. As aforementioned, the main challenges and bottlenecks of the problem are 1) sparse and incomplete observation, and 2) the lack of a suitable stochastic traffic flow model. To address these issues, we propose a new stochastic traffic flow model and utilize it to construct the overall traffic flow with low penetration rate vehicle trajectory data. The calibrated stochastic traffic flow model enables different kinds of traffic signal optimization approaches. In this paper, we propose relatively simple but effective methods to generate new signal timing plans. This is designed for implementation purposes since interpretability, reliability, and robustness is more important.

We agree with the reviewer that there are different methods to generate new signal timing plans, including different optimization-based methods and data-driven methods like RL. Although this is not the focus of this study, we have provided a literature review in the supplementary materials. Please refer to Section 2.3 in Supplementary Information for the full content, where we have introduced different methods and commented on the associated limitations. Here is the paragraph related to RL-based methods:

“Reinforcement Learning (RL) has become a popular approach for traffic signal control, as evidenced by several studies⁸⁻¹³. RL can directly learn an end-to-end control policy from the observation by interacting with the simulation environment iteratively. Most of the existing literature using RL for traffic signal control focuses on the design of the input state space and reward¹² while utilizing different RL techniques such as the multi-agent algorithms^{11,13}. Despite the abundance of research utilizing RL for traffic signal optimization, there remains a significant gap between research and implementation in the real world. One of the main concerns is the reliability of RL-based approaches. RL controllers trained offline in a simulation environment may not perform well in real-world scenarios due to the limited fidelity of the

simulation. On the other hand, training RL controllers directly in the real world raises additional challenges, particularly in managing the risks associated with exploration during the learning process.”

6. *Some works using computer vision to estimate the traffic volume and adjust the signals in real-time. Please compare the advantages and disadvantages.*

Response: Computer-vision-based methods require cameras installed at the signalized intersection and are considered as detector-based methods. We have provided the comparison between detector data with vehicle trajectories in Line 43, Page 1:

“Monitoring traffic through vehicle trajectory data offers many advantages over fixed-location detectors and sensors. It has a much larger coverage area than detector data because it is available at almost every intersection, especially those with higher traffic volumes (Fig. 1a). While detector data can only provide traffic counts and estimated speeds at certain locations, vehicle trajectory data spans the entire spatial-temporal space and provides more enriched information such as delay, number of stops, and travel path (Fig. 1b).”

In this study, we propose to utilize vehicle trajectory data without relying on any road-side detectors. The advantage of detector data is that it can provide the complete traffic volumes at installed locations, however, it has limited installation. Vehicle trajectory data, on the other side, is available at almost every intersection but limited by the low penetration rate. The incomplete observation caused by a low penetration rate limits the performance of a more responsive control. We have added the following to the Discussion section (Line 391, Page 12):

“However, the incomplete observation caused by low penetration rates may limit the accuracy of the real-time traffic state estimation. This will be improved in the future when more vehicles are connected.”

7. *One of the research trends is signal-free intersection (E.g., DOIs: 10.1109/MITS.2017.2743167, 10.1109/ACCESS.2018.2871337). It is worth discussing.*

Response: Please also refer to our response to Question 1 of Reviewer #2. A signal-free intersection usually requires that all vehicles are automated, which is not the focus of this study and might not be feasible in the near future. This paper focuses on traffic signal re-timing technique with currently available connected vehicle data so that it can be directly implemented in the field.

Reviewer #3 (Remarks to the Author)

The manuscript presents a new traffic signal timing system that uses connected vehicle data. They first connect a stochastic point-queue model with vehicle trajectories under a proposed Newellian (inspired by Newell’s car-following behavior) coordinates to construct spatiotemporal traffic states. They then use an optimization algorithm to update traffic signal parameters. They implemented signal optimization on a real-world network in Michigan and observed improvements in traffic delay and the number of stops using their proposed framework. Overall, the manuscript is very well-written, covers an intriguing topic, and presents a novel and valid methodology.

My comments/suggestions are as below.

1. *It is stated in the manuscript, for instance, that “we present the world’s first large-scale cloud based traffic signal re-timing system that uses a small percentage of connected vehicle trajectories as the only input without reliance on any infrastructure-based detectors.”, “By utilizing the trajectory data as the only input and not requiring any additional infrastructure, OSaaS provides a more scalable and economical solution to traffic signal retiming which can potentially be applied to every fixed-time traffic signal in the world.”, or “We present the world’s first large-scale cloud-based traffic signal optimization system (OSaaS: Optimizing Signals as a Service) based on vehicle trajectory data collected by connected vehicle service providers, which could be independent of traffic management agencies.” These sentences imply that this study is the first one that optimizes signal timing using vehicle trajectory data. However, there are many other studies in the literature that utilize only this data and not detectors’ data for signal optimization. In order to provide a comprehensive evaluation of the manuscript’s contributions, I would recommend discussing its unique aspects in relation to these existing studies. Emphasizing the distinctive contributions on these points would enhance the manuscript’s overall clarity and significance. For your reference, I have included two examples of studies that utilize vehicle trajectory data below:*

- 1) *Xu, Biao, Xuegang Jeff Ban, Yougang Bian, Wan Li, Jianqiang Wang, Shengbo Eben Li, and Keqiang Li. "Cooperative method of traffic signal optimization and speed control of connected vehicles at isolated intersections." IEEE Transactions on Intelligent Transportation Systems 20, no. 4 (2018): 1390-1403.*
- 2) *Li, Wan, and Xuegang Ban. "Connected vehicles based traffic signal timing optimization." IEEE Transactions on Intelligent Transportation Systems 20, no. 12 (2018): 4354-4366.*

Response: We agree with the reviewer that this is not the only study that proposes to utilize the vehicle trajectory data for traffic signal optimization, and we have changed the main paper and avoid using of “the world’s first...”. However, to the best of the authors’ knowledge, few studies have been implemented in the field with the scale of this paper. This work used real-world data and performed signal retiming for all 32 intersections in the City of Birmingham in Michigan.

We have revised the Introduction section in the main paper to further clarify the contribution of this paper. The related paragraph is written as (Line 69, Page 2):

“This paper focuses on the optimization of fixed-time traffic signals using connected vehicle trajectories and does not rely upon any road-side detectors (e.g., loop detectors, cameras). A vehicle is considered to be a connected vehicle if its location is known such that its trajectory can be constructed. Although many existing studies have investigated traffic signal control with connected and automated vehicles (CAV) I-7, they assume a high penetration rate (i.e., the proportion of CAVs to the overall number of vehicles), which is not realistic in the current practice. In this study, we aim at optimizing traffic signals utilizing vehicle trajectories at the currently available market penetration rate. In this case, one major challenge is the resulting sparse and incomplete observation of the overall traffic state. Some studies have developed statistical methods to estimate certain traffic flow parameters such as traffic volumes or queue lengths, but they can only be used for traffic monitoring purposes due to the lack of an explicit traffic flow model. For traffic signal optimization, it is important to have the capability to predict traffic flow performance under different traffic signal parameters.”

We believe this added paragraph highlights the unique aspects of our work. The main challenges of the problem are twofold: 1) incomplete and sparse observation; and 2) the lack of a suitable stochastic traffic flow model. Therefore, while many of these studies focused on the optimization methods (or other methods such as reinforcement learning mentioned by Reviewer #2), we focus more on establishing a well-calibrated stochastic traffic flow model to reconstruct the overall traffic state (system identification) with a realistic penetration rate.

Specifically, we propose a new stochastic traffic flow model with lower dimensions that can be directly calibrated with vehicle trajectory data. We also apply the newly proposed probabilistic time-space (PTS) diagram to obtain the spatial-temporal distribution of vehicle trajectories.

2. *The manuscript utilized connected vehicle data from General Motors (GM) for the case study. The usage of the term 'connected vehicle' might be slightly confusing in this context since GM vehicles are not connected to each other or an infrastructure. To the best of my knowledge, the GM data consists of GPS data from these vehicles, which can be considered partially connected. I would suggest clarifying the definition of connected vehicles and providing more specific details regarding the data.*

Response: We agree with the reviewer that according to a more rigorous definition, the GM vehicle is partially connected since it can only provide information to the cloud. In this paper, we use a broader definition (Line 70, Page 2): “A vehicle is considered to be a connected vehicle if its location is known such that its trajectory can be constructed.” We have added this definition to the main paper to avoid confusion.

The specification of the data is available in the Methods section, it is written as (Line 529, Page 19):

“The vehicle trajectory data in this work is from General Motors (GM) vehicles, which are equipped with GNSS (Global Navigation Satellite System) receivers and inertial measurement units (IMUs) that provide accurate vehicle position and dynamics information. These vehicles also have wireless communication capability (5G, LTE etc.) and support quick communication with cloud services. As a result, the vehicles can act as real-time mobile sensors that enable smart traffic signal operations. Trajectory point attributes include a unique trip ID, GNSS coordinates (latitude and longitude), timestamp, and speed. Their accuracy is roughly within 3-5 meters, and they are received at a time interval of approximately 3 seconds. For the studied area (City of Birmingham, Michigan, US), there are approximately 2 million points and 25 thousand unique trips each day. The penetration rate is estimated to be around 7% according to this study.

The road network in this study is re-organized from OpenStreetMap, which is open-source and available online. Trajectories are matched to the road network so that we can convert raw GNSS coordinates to distance information of certain road segments. Existing signal phase and timing (SPaT) data is extracted from signal work orders provided by the Road Commission for Oakland County.”

We also have an illustration video in Supplementary Movie 1, which visualizes raw vehicle trajectory data.

References

1. L. Chen and C. Englund, "Cooperative intersection management: A survey," *IEEE transactions on intelligent transportation systems*, vol. 17, p. 570–586, 2015.
2. J. Li, C. Yu, Z. Shen, Z. Su and W. Ma, "A survey on urban traffic control under mixed traffic environment with connected automated vehicles," *Transportation Research Part C: Emerging Technologies*, vol. 154, p. 104258, 2023.
3. W. Li and X. Ban, "Connected vehicles based traffic signal timing optimization," *IEEE Transactions on Intelligent Transportation Systems*, vol. 20, p. 4354–4366, 2018.
4. B. Xu, X. J. Ban, Y. Bian, W. Li, J. Wang, S. E. Li and K. Li, "Cooperative method of traffic signal optimization and speed control of connected vehicles at isolated intersections," *IEEE Transactions on Intelligent Transportation Systems*, vol. 20, p. 1390–1403, 2018.

5. Q. Guo, L. Li and X. J. Ban, "Urban traffic signal control with connected and automated vehicles: A survey," *Transportation research part C: emerging technologies*, vol. 101, p. 313–334, 2019.
6. P. Lin, J. Liu, P. J. Jin and B. Ran, "Autonomous vehicle-intersection coordination method in a connected vehicle environment," *IEEE Intelligent Transportation Systems Magazine*, vol. 9, p. 37–47, 2017.
7. Y. Feng, K. L. Head, S. Khoshmagham and M. Zamanipour, "A real-time adaptive signal control in a connected vehicle environment," *Transportation Research Part C: Emerging Technologies*, vol. 55, p. 460–473, 2015.
8. Arel, I., Liu, C., Urbanik, T. and Kohls, A.G., 2010. Reinforcement learning-based multi-agent system for network traffic signal control. *IET Intelligent Transport Systems*, 4(2), pp.128-135.
9. Khamis, M.A. and Gomaa, W., 2014. Adaptive multi-objective reinforcement learning with hybrid exploration for traffic signal control based on cooperative multi-agent framework. *Engineering Applications of Artificial Intelligence*, 29, pp.134-151.
10. Yau, K.L.A., Qadir, J., Khoo, H.L., Ling, M.H. and Komisarczuk, P., 2017. A survey on reinforcement learning models and algorithms for traffic signal control. *ACM Computing Surveys (CSUR)*, 50(3), pp.1-38.
11. Chu, T., Wang, J., Codecà, L. and Li, Z., 2019. Multi-agent deep reinforcement learning for large-scale traffic signal control. *IEEE Transactions on Intelligent Transportation Systems*, 21(3), pp.1086-1095.
12. Wei, H., Zheng, G., Gayah, V. and Li, Z., 2019. A survey on traffic signal control methods. *arXiv preprint arXiv:1904.08117*.
13. Wu, Q., Wu, J., Shen, J., Du, B., Telikani, A., Fahmideh, M. and Liang, C., 2022. Distributed agent-based deep reinforcement learning for large scale traffic signal control. *Knowledge-Based Systems*, 241, p.108304.

REVIEWERS' COMMENTS

Reviewer #2 (Remarks to the Author):

The authors have addressed my comments, and I do not have any additional remarks.

Reviewer #3 (Remarks to the Author):

The authors have addressed all my comments. I find the paper interesting and I believe it has sufficient contributions for publication in the Nature Communications journal.

Additional note: Unfortunately, Reviewer #1 was unavailable to provide a review for the revised manuscript. To understand whether the remaining concerns raised by Reviewer #1 from the previous round of review would prevent publication or not, we consulted Reviewer #3 with a set of corresponding questions to these concerns (Reviewer #1's answers follow each question below). As you will see, Reviewer #1 confirms that nearly all of the concerns have been suitably addressed. However, based on their comments, we strongly recommend that you consider simplifying the writing in the main text for overall clarity and more clearly highlighting and presenting the shockwave validation method provided in your revised manuscript.

1) Are the mathematical formulations in this work clearly presented?

The paper is still hard to read, and it is a bit challenging to follow the mathematical formulations and connect them to each other. A part of the complexity might be from the paper covering various aspects of traffic flow and network analysis, making it both complicated and interesting. However, I believe the writing could be improved and simplified.

2) Is the probabilistic space-time diagram clearly presented?

I believe it has been sufficiently explained.

3) Is the stochasticity in the model clearly presented?

The authors have used a stochastic point-queue model, which has been clearly explained.

4) Do the Methods section and Supplementary Materials Section 7 present a clear explanation of signal timing, offset and phasing optimization?

The paragraph added to the Methods section sufficiently explains the signal timing optimization method of this study. The supplementary materials section is more of a literature review, which is also useful.

5) Do the Split-failure ratios in Supplementary Table 1 provide a clear indication of the intersection saturation levels?

Split-failure ratio can be used as an indication of intersection saturation level, so I believe the author's explanation is clear.

6) Are the shockwave properties and shockwave validation clearly explained?

I do not quite understand the reviewers' comment on shockwave validation. If it means validating it with empirical data, no explanation or clarification is included in the paper.

7) The authors have stated that they have not included measures of queue length, intersection capacity and discharge profile as they do not form part of the model's objective function and can be estimated from the queueing model developed. Do you feel that these measures should necessarily be presented, or do they fall outside the scope of the presented work as the authors indicate?

I agree with the authors that these measures can be estimated using the common methods in the literature.

Response to the Reviewers' Comments

The authors would like to thank the editor and reviewers for their constructive comments. In response, we have carefully revised the paper and highlighted our changes in red. We believe the revised paper addresses the remaining issues and concerns raised by the reviewers. The following is a point-to-point response.

Are the mathematical formulations in this work clearly presented?

The paper is still hard to read, and it is a bit challenging to follow the mathematical formulations and connect them to each other. A part of the complexity might be from the paper covering various aspects of traffic flow and network analysis, making it both complicated and interesting. However, I believe the writing could be improved and simplified.

Response: We made further changes to the main paper to improve its readability, particularly the mathematical formulation. One of the main reasons that it is hard to read might be that each figure in the previous submission contains too much information so that it would be hard to understand. Therefore, we re-organized Fig 1c and Fig 2a in the previous submission and split them into three new figures: the current Fig 1c, Fig 2a and Fig 2b. We also changed the description in the main paper to reflect these changes. We believe that the new figures as well as the description are more logical and clearer: Fig 1c (new) shows traditional traffic state representations including Eulerian and Lagrangian. Fig 2a (new) shows how the proposed Newellian coordinates are established while Fig 2b (new) shows how the vehicle trajectories are converted to a point-queue representation in the Newellian coordinates.

Original Fig 1c

Original Fig 2a

New Fig1 c

New Fig 2a and b

Besides, we also have Supplementary Movie 2 & 3, which we believe would be helpful for readers to understand how the proposed model works.

Are the shockwave properties and shockwave validation clearly explained?

I do not quite understand the reviewers' comment on shockwave validation. If it means validating it with empirical data, no explanation or clarification is included in the paper.

Response: Fig 4e-f in the main paper illustrates how we validate the shockwave generated through the proposed probabilistic time-space diagram with real-world observations. We have further clarified this in the main paper (Line 222):

“...The spatial-temporal space can be separated into queueing area and free-flow area according to a pre-determined speed threshold. White dashed lines are the boundary between queueing area (dark color) and free-flow area (light color). These boundary lines are also referred to as shockwaves in traffic flow theory, which separate the spatial-temporal traffic into different areas with relatively uniform traffic states. To validate the reconstructed traffic state as well as shockwaves, we use IoU (Intersection over Union) of the queueing area to quantify the similarity between ground truth and reconstructed heatmaps....”